# Molecular and Hormonal Mechanisms Regulating Fleshy Fruit Ripening

**DOI:** 10.3390/cells10051136

**Published:** 2021-05-08

**Authors:** Shan Li, Kunsong Chen, Donald Grierson

**Affiliations:** 1College of Agriculture & Biotechnology, Zijingang Campus, Zhejiang University, Hangzhou 310058, China; lishan-987@163.com; 2Zhejiang Provincial Key Laboratory of Horticultural Plant Integrative Biology, Zijingang Campus, Zhejiang University, Hangzhou 310058, China; 3Plant and Crop Sciences Division, School of Biosciences, Sutton Bonington Campus, University of Nottingham, Loughborough LE12 5RD, UK

**Keywords:** colour, ethylene, flavour, fruit ripening, softening, plant hormone, transcription factor, epigenetics

## Abstract

This article focuses on the molecular and hormonal mechanisms underlying the control of fleshy fruit ripening and quality. Recent research on tomato shows that ethylene, acting through transcription factors, is responsible for the initiation of tomato ripening. Several other hormones, including abscisic acid (ABA), jasmonic acid (JA) and brassinosteroids (BR), promote ripening by upregulating ethylene biosynthesis genes in different fruits. Changes to histone marks and DNA methylation are associated with the activation of ripening genes and are necessary for ripening initiation. Light, detected by different photoreceptors and operating through ELONGATED HYPOCOTYL 5(HY5), also modulates ripening. Re-evaluation of the roles of ‘master regulators’ indicates that *MADS-RIN*, *NAC-NOR*, *Nor-like1* and other *MADS* and *NAC* genes, together with ethylene, promote the full expression of genes required for further ethylene synthesis and change in colour, flavour, texture and progression of ripening. Several different types of non-coding RNAs are involved in regulating expression of ripening genes, but further clarification of their diverse mechanisms of action is required. We discuss a model that integrates the main hormonal and genetic regulatory interactions governing the ripening of tomato fruit and consider variations in ripening regulatory circuits that operate in other fruits.

## 1. Introduction

Fruits can develop from several different parts of a flower and possess different characteristics. In this review, we shall consider the biochemical, genetic and hormonal mechanisms involved in the ripening of fleshy fruits, which makes them attractive to frugivores (animals that consume raw fruits, succulent fruit-like vegetables, roots, shoots, nuts and seeds). When the seeds are mature, the cells in the fruit flesh undergo dramatic changes in colour, flavour, texture and aroma that attract frugivores to aid in seed dispersal [1]. This ripening process involves a coordinated set of physiological and biochemical changes involving many cell compartments, including the nucleus (Section 4, Section 5 and Section 6), plastids and vacuole (Section 2.1) and cell wall (Section 2.2). Ripening is under genetic control and has evolved several times during the course of evolution [2]. The ripening process depends upon the expression of new ‘ripening genes,’ and the enzymes they encode catalyse a range of biochemical changes that render the fruits attractive to humans and other seed-dispersing animals. Different structural genes and control mechanisms have been recruited to the ripening program in different fruit during the course of their evolution. Fleshy fruit ripening is of great scientific interest because fruits make an important contribution to animal and human diets, provide several health-promoting benefits [3] and fruit production underpins major economic activity worldwide.

It is generally accepted that fleshy fruits can be divided into two classes, the climacteric fruits (i.e., apple, apricot, avocado, banana, tomato, etc.) and non-climacteric fruits (i.e., citrus, grape, orange, lemon, raspberry, strawberry, etc.). The main differences between these two groups are the characteristic burst of respiration and ethylene production that occurs at the onset of ripening in climacteric fruits [3,4,5]. Non-climacteric fruits respire and produce low levels of ethylene, and they can also sometimes respond to this hormone, which promotes, for example, de-greening of citrus. Non-climacteric fruits do not, however, show a burst of CO_2_ or ethylene production, which is the classic sign of ripening onset in climacteric fruit. This distinction between two types of fruits is somewhat arbitrary, however, and ripening of both types of fruits involves hormones and similar changes in gene expression, often regulated by structurally related transcription factors (TFs). Hormone signalling operates primarily by modulating expression or action of TFs plus genes involved in hormone biosynthesis, perception and signalling. TFs can also directly and indirectly affect expression of genes required for hormone synthesis, and expression of some ripening TFs is regulated developmentally or by environmental stimuli. Consequently, there is a balance between developmental, environmental, hormonal and genetic cues that govern the processes of fruit development and ripening.

The tomato (*Solanum lycopersicum*) was selected as a model for fruit ripening and the complete genome sequence was determined in 2012 [6]. The tomato genome contains 58 TF families, including a total of 1845 putative TFs [7]. For a complete understanding of the ripening process, it is necessary to know how these TFs and different hormones and environmental factors operate and interact at the molecular level to coordinate the myriad of ripening reactions. Although the genetic and biochemical changes that occur during ripening are important determinants of fruit quality, if some aspects of ripening, such as softening, proceed too far, the fruit become susceptible to infection, spoilage and rotting. Thus, understanding how the different facets of ripening are regulated can help improve fruit quality and quantity, prevent waste and contribute to improved nutrition and food security.

Although tomato has been an important ripening model [3,8,9,10,11,12], the fact that ripening has arisen several times during the course of evolution, and the differences between climacteric and non-climacteric fruits, caution against a “one type fits all” model [13]. Furthermore, initial conclusions about the function of key tomato ripening “master regulator” TFs (NAC-NOR, MADS-RIN, SPL-CNR), which strongly influenced ripening models, have turned out to be misleading and their roles in ripening have been revised [14,15,16,17,18,19,20]. In this article, we shall focus on the recent, very significant changes in understanding of the mechanisms underlying the control of fleshy fruit ripening and quality. We consider important information garnered from research on a range of fruits, where this has enhanced our understanding of ripening, and develop a model of how the different aspects of the ripening processes are integrated.

## 2. Main Physiological and Biochemical Changes Are Regulated by Ripening Genes

Massive technological progress has accelerated the accumulation and analysis of data in the last 15 years. Complete genomic sequences of hundreds of angiosperms are now available, including over 90 fruit species of economic importance (https://simple.wikipedia.org/wiki/List_of_fruits, accessed on May 6 2021) and other horticultural crops [21,22], and the number is rapidly increasing. With the advent of RNA-seq technology, transcriptomic sequences for many developmental processes, including ripening, are rapidly becoming available. Where it once took 2–3 years to identify and characterise the structure and expression of a single gene, it is now possible to acquire this information for 25,000 genes in a few weeks, and, unlike the early years, clues to gene function can often be gained from DNA sequence homology searches against previously characterised genes. Many ripening genes affecting different aspects of fruit physiology and biochemistry have been identified by this approach [23]. Although this impressive progress has contributed large amounts of data (Sol Genomics Network, https://solgenomics.net/; Tomato Expression Atlas, http://tea.sgn.cornell.edu/, accessed on 6 May 2021), there is still a need to consider it in a physiological and biochemical context, in order to understand its significance and unravel the hormonal, genetic and biochemical regulatory mechanisms involved in ripening.

### 2.1. Colour Changes during Ripening

Visual clues are extremely important for signalling at a distance to frugivores the availability of ripe fruit. A wide range of animals, including seed-dispersers such as bats, rodents, primates and many different types of birds, with different vision systems use visible (380 to 700 nm) or UV light (mainly UVA, 315 to 400 nm) to detect them. Different organisms can have between 1 and 4 functioning colour receptors. Many mammals, including bats and rodents, are dichromats (i.e., have only two types of colour receptor), yet are capable of distinguishing ripe fruits from leaves. Most new world monkeys are also dichromats, whereas most other primates have three types of colour receptors (i.e., they are trichromats), which confer improved colour discrimination.

There are four main mechanisms involved in colour change in fruits:(1)Loss of chlorophyll from chloroplasts, which is accompanied by the disassembly and recycling of some thylakoid membranes and photosynthetic proteins.(2)Accumulation of coloured carotenoids such as β-carotene and lycopene in lipid globules or other characteristic membrane-bound structures in the developing chromoplasts, which arise either by conversion of chloroplasts or develop from other types of plastids [24,25,26,27].(3)The accumulation of flavonoids or anthocyanins pigments in the cell vacuoles.(4)The production of UV-reflecting components such as surface waxy layers, fatty deposits within cells or from parallel sheets of membranes that generate iridescence due to thin film interferences of light [28].

The synthesis of anthocyanins and carotenoids during ripening has been the most intensively studied at the molecular level and involves the selective expression of genes encoding specific enzymes in their respective metabolic pathways.

Much of the knowledge on the regulation of carotenoid biosynthesis has been obtained from tomato and can be inferred to apply to other fleshly fruits (Figure 1), although it should be emphasised that structurally different carotenoids can occur in different types of fruits. Fruits rich in carotenoids provide a rich source of dietary pro-vitamin A [3,29]. The precursor of carotenoid biosynthesis, geranylgeranyl pyrophosphate (GGPP), is synthesised via the methylerythritol-4-phosphate (MEP) pathway. Two GGPP molecules produce phytoene, catalysed by phytoene synthase (PSY). Phytoene is converted into lycopene by undergoing a sequential series of desaturation and isomerisation reactions by phytoene desaturase (PDS), ζ-carotene isomerase (ZISO), ζ-carotene desaturase (ZDS) and carotene isomerase (CRTISO). Lycopene can be cyclised by lycopene cyclase (LCY) to form α-carotene, generating lutein, catalysed by carotene hydroxylase (CHX). Alternatively, lycopene can be cyclised by LCY or chromoplast-specific lycopene β-cyclase (CYCB) to form β-carotene, which is converted into β-cryptoxanthin and then into zeaxanthin by CHX [30]. Tomato PSY1 was the first carotenoid biosynthetic enzyme to be identified [31] and the pathways and enzymic steps for carotenoid production are now well-established [30]. In tomato, the PSY1 isoform is involved in synthesis of carotenoids in ripening fruit, whereas PSY2 performs this function in green tissues. Several other key genes are also upregulated during ripening [32,33] and there is also substantial variation between fruits. A recent review of mutations underlying colour differences between different fruits and varieties highlighted mutations affecting coding sequences and promoters of a number of carotenoid biosynthetic enzymes, including CRTISO, ZISO, CHX, CYCB, etc. [30,34].

Anthocyanins are water-soluble, purple and red pigments (cyanidin, pelargonidin and delphinidin derivatives) that accumulate in the vacuoles of epidermal and/or flesh cells of fruits, as they ripen. They are synthesised from precursors generated from the upper part of the phenylpropanoid pathway (not shown). Many fruits are enriched with anthocyanins, such as cyanindin-3-glucoside (C3G), which is the predominant anthocyanin in Chinese bayberry [35], mulberry, apple peel [36,37,38] and peach skin [39], and other predominant anthocyanins such as pelargonidin in strawberry [40], cyanidin and delphinidin in grape skin [41] and cyanidin-3-rutinoside in cherry [42].

Transcriptional regulation of the downstream structural genes (such as chalcone synthase (CHS), chalcone isomerase (CHI), flavanone 3-hydroxylase (F3H), flavonoid 3′-hydroxylase (F3′H), flavonol synthase (FLS), dihydroflavonol 4-reductase (DFR), flavone 3′-*O*-methyltransferase 1 (OMT1) and anthocyanidin synthase (ANS)) governs the three main branches that lead to the production of the different flavonoid pigments in many plant species [43] (Figure 2).

Expression of many of the anthocyanin biosynthetic genes is regulated by a conserved mechanism involving a complex comprised of three types of TFs: MYB + bHLH (basic helix–loop–helix) + WD40 (WD40 repeat protein) (MBW) [45,46]. There are many different types of MYBs, including transcriptional activators, and some, such as the R3-MYB protein MYBL2 or the R2R3-MYB protein MYB27, possess a C-terminal EAR motif and act as repressors [47].

External colour change mechanisms are not universal, however, and ripe ‘Hayward’ kiwifruit remain green, or greenish brown externally and light green internally, whereas ‘Hort16A’ kiwifruit turn gold internally [48]. Fruit inner pericarp of *Actinidia chinensis* cv. Hort22D has red flesh, because AcMYBF110 is thought to activate the transcription of DFR, ANS and F3GT1 [49]. In addition, upregulation of *MYBA1-1* and *MYB5-1* by low temperature enhances anthocyanin accumulation in ‘Hongyang’ kiwifruit by transcriptional activation of ANS1, ANS2, DFR1, DFR2 and UFGT2 [50]. Also, different mechanisms are not mutually exclusive, and several can operate in the same fruit. For example, during ripening of some apples, the chloroplasts in the peel cells lose chlorophyll and accumulate carotenoids, and sometimes also anthocyanins, although some varieties remain green [38,51]. Tomatoes normally lose chlorophyll and accumulate carotenoids, but “*greenflesh*” mutant fruit retain some thylakoids and chlorophylls, and at the same time, they accumulate carotenoids, and consequently, the fruit appear a dirty-brown colour [52]. Furthermore, tomatoes can also accumulate anthocyanins as well as carotenoids if they are manipulated to express the necessary genes [53]. Strawberries, black grapes and Chinese bayberry are examples of fruits that lose chlorophyll and normally only accumulate anthocyanins [40,54,55].

### 2.2. Fruit Softening

It is generally accepted that fruit softening during ripening is, at least partly, due to cell wall changes catalysed by wall-modifying enzymes. Investigations with many fruits have shown that there are at least 10 different types of cell wall-modifying enzymes that are expressed during ripening [56], and these are believed to collectively contribute to the change in texture, leading to alterations in the structure and hydration of cell wall polymers, leading to softening. These include the pectin-modifying enzymes: pectinesterase (PE), pectate lyase (PL), polygalacturonase (PG) and β-galactosidase (β-GAL), and the hemicellulose/cellulose-modifying enzymes: 1,4-β-glucanase, xyloglucan transglycosylase/hydrolase (XTH) and expansin (EXP). It is clear that several of these enzymes have specific actions affecting cell wall structure that contribute to texture changes, although there is no single enzyme that makes a major contribution to softening [56,57]. Expansins allow cellulose microfibrils to slide over one another without covalent modification, and PG reduces the chain length of pectins, which does have some effect because it decreases the viscosity of tomato extracts [58]. Softening of intact fruit is largely unaffected, although some small increase in firmness has been detected in antisense-PG tomatoes [59]. Furthermore, an analysis of *PG*/*EXP1* knockdown tomato fruit showed that they were firmer, had higher total soluble solids, denser cell walls and thicker cuticles than fruit of other genotypes [60], and also showed reduced skin cracking [60]. More recently, RNAi knockdown and CRISPR/Cas9 knockout experiments showed that eliminating *PL* produced firmer tomatoes, although softening was not abolished [61].

The question of the regulation of cell wall metabolising enzymes was addressed early on and there are differences between fruits. *PG* transcripts in tomato are very sensitive to low levels of ethylene [62]. The situation varies in different fruits, however, and in melon, which expresses three *PG* genes, accumulation of *CmPG1* transcripts is dependent on ethylene, whereas regulation of *CmPG2* is ethylene-independent and expression of *CmPG3* is regulated by both ethylene-dependent and ethylene-independent factors [63,64].

Attempts to delay fruit softening and improve fruit quality by manipulating stress responses during storage have provided new insights into the role of a group of cell wall-modifying enzymes in causing texture change. Deterioration of peaches is delayed if they are stored at 0 °C. Upon removal from storage, however, they fail to undergo normal softening, and consequently, the texture is disliked by consumers due to the failure to produce the normal levels of a group of cell wall-modifying enzymes. A low-temperature conditioning (LTC) treatment at 8 °C, however, prior to 0 °C storage, results in much improved softening upon return to room temperature after 0 °C storage. The explanation for this is that after transfer to room temperature, transcripts for *PG*, *PL*, *PE*, *β-1,4-endoglucanase*, *xylosidase* (*XYL*), *EXP1*, *GAL* and *XTH* are all substantially increased, provided the peaches were acclimated at 8 °C prior to 0 °C storage [65]. Similarly, in persimmon fruit, many cultivars are astringent, due to the high soluble tannins content, which consumers find unacceptable. There are several postharvest treatment methods for removing these soluble tannins. One successful approach is to store the fruits in a 95% CO_2_ + 1% O_2_ atmosphere. This postharvest high CO_2_/anaerobic storage leads to acetaldehyde production, which precipitates the soluble tannins, reducing or eliminating the astringency. Unfortunately, however, persimmon treated in this way tend to soften too much. It was found that adding the ethylene perception inhibitor 1-MCP to the 95% CO_2_ + 1% O_2_ storage atmosphere greatly reduced the appearance of ethylene response factor TFs (ERFs) 8, 18 and 19, that upregulate a similar group of genes as those in peach, which together are responsible for modifying cell walls in persimmon fruit [66]. This indicates that ethylene is responsible for stimulating persimmon softening. Significantly, the promoters of the cell wall genes in peach also have ERF binding sites [65], suggesting that in both of these cases, ethylene may be involved in regulating several different genes that modify cell wall structure and softening [56]. The situation in non-climacteric fruits is less clear. Villarreal et al. [67] provided evidence that ethephon (which produces ethylene in treated plants) and 1-MCP treatment (which inhibits ethylene perception and action) influenced the expression of both *PG* and *β-Gal*, although Nardi et al. [68] showed that accumulation of *FaEXP2* in strawberry fruit, whilst appearing to be influenced by both ABA and auxin, was unaffected by either ethylene or 1-MCP.

### 2.3. Synthesis and Accumulation of Organic Acids, Sugars and Volatiles Contribute to Taste and Aroma

The main taste characteristics of fruit are derived from the relative concentrations of organic acids and sugars [69]. The most common sugars are sucrose, glucose and fructose, which are ultimately derived from photosynthate imported during development. This soluble carbohydrate is often converted to starch and stored in the plastids of unripe fruit. During ripening, the starch is initially metabolised to maltose and glucose, by a network of starch-degrading and sugar-metabolising enzymes. During development, bananas accumulate 20–25% fresh weight of starch, which is converted to sugars during ripening. In some other fruits, photosynthesis by fruit chloroplasts can also contribute to the accumulation of stored starch prior to ripening. Eighteen starch degradation-associated enzymes from banana have been identified bound to the surface of starch granules, and ten showed increased activity during fruit ripening. The accumulation of transcripts and enzyme activities of some of these genes are ethylene inducible. A transcriptional activator MabHLH6 (a basic helix–loop–helix TF) preferentially binds to the promoters of 11 genes encoding starch degradation enzymes and transactivates their promoters [70]. Other types of TFs are also involved in regulating expression of starch degradation genes, since a zinc finger TF, DNA BINDING WITH ONE FINGER (AdDof3), physically interacts with the promoter and transactivates the *AdBAM3L* (β-amylase) gene in ripening kiwifruit [71]. The metabolism of the large amount of starch in banana not only generates sugars during ripening but also contributes directly to a reduction in firmness, in addition to cell wall changes outlined in Section 2.2. Sugar accumulation often involves the activities of sucrose synthase and sucrose phosphate synthase [72], and appropriate transporters [73] and increases in acid and neutral invertases during ripening of tomato convert sucrose to glucose and fructose, which enhances sweetness [74]. In watermelon, four genes involved in the metabolism of sugars have been identified as being expressed in different watermelon genotypes, including three sugar transporters, SWEET, EDR6 and STP, two sucrose phosphate synthases and sucrose synthase genes [75].

Unlike sugars, which are imported to the fruit as photosynthate, organic acids are generally synthesised in situ, mostly from sugars, metabolised starch and products of cell wall metabolism [63]. Malate and citrate are the most common fruit organic acids that accumulate in both climacteric and non-climacteric fruits during fruit development, although other acids are found in some fruits. In watermelon, three genes involved in the TCA cycle and pH regulation have been identified that are believed to play a role in organic acid accumulation [75].

During ripening, organic acids can slowly decrease in concentration in some fruits, as the malate, and sometimes citrate, are utilised as respiratory substrates, whereas sugar contents generally continue to increase. It is possible that these sugars and organic acid may have another role, since sugars are known to be signalling molecules as well as being a form of soluble carbohydrate [76]. Furthermore, when malate metabolism in tomato was modified by antisense inhibition of the activities of mitochondrial malate dehydrogenase (MDH), it was found that lines with low malate showed an increased accumulation of starch. Fruit softening and the content of chlorophylls, carotenoids and other pigments were also affected, and the low-MDH tomatoes showed increased susceptibility to *Botrytis cinerea* infection. It is possible that some of these effects are due to the altered redox state due to lowered malate, leading to several biochemical changes [77].

The characteristic flavour and aroma notes of different fruits are conferred by a range of different volatile components that are synthesised from several different biochemical pathways. Lipoxygenase (LOX) and alcohol dehydrogenase (ADH) in tomato [78,79] and alcohol acyl-transferase (AAT) in melon [80] were among the first enzymes to be identified that were associated with production of flavour and aroma volatiles during ripening. Identifying the correct LOX in tomato turned out to be complicated because there were at least five different isoforms involved in different biochemical pathways, including the production of jasmonic acid (JA), which will be discussed later. Antisense technology, using targeting sequences specific for each gene, identified the plastid-located TOMLOX C as playing a critical role in production of lipid-derived C6, and later, C5 volatiles [13]. Alcohol acetyltransferase (AAT1) is important in volatile ester synthesis [80,81] and the final step in the pathway, conversion of aldehydes to alcohols, requires ADH2 activity [79]. Recently, an increase in FaLOX5 transcripts has been correlated with volatile esters’ production in ripening strawberry, and FaMYB11 promoted volatile accumulation partly through the transcriptional activation of FaLOX5 [82]. Many other genes are now known that contribute to the production of flavour volatiles, and it is important to recognise that they are derived from several different biochemical pathways, including fatty acids, amino acids and carotenoids (Figure 3). Key genes that contribute to aroma and flavour in ripe tomato fruits include branched-chain amino acid aminotransferases (*BCAT1*), which is important in the production of branched-chain amino acids [83,84,85]. A major group of amino acid-derived flavour and aroma compounds, benzenoids (C6–C1), are synthesised from L-phenylalanine, including benzaldehyde and benzyl alcohol. The first step in this biosynthetic pathway is catalysed by L-phenylalanine ammonia lyase (PAL), encoded by the *PAL3* gene, which converts phenylalanine to trans-cinnamic acid [86]. Aromatic amino acid decarboxylase (AADC1A) mediates the first step in the production of phenylalanine-derived volatiles in tomato fruits [87], and the transcripts of the carotenoid cleavage dioxygenase 1 (*CCD1B*) gene also participate in aroma volatiles from the amino acid pathway.

Terpenoids constitute the largest group of plant volatile secondary metabolites [89,90] and function in attraction and defence during vegetative and reproductive growth. They are synthesised from isopentenyl diphosphate (IPP) and dimethylallyl diphosphate (DMAPP) via two pathways: the cytosol-located mevalonate (MVA) pathway and the methyl-erythritol phosphate (MEP) pathway in the plastid [91]. Terpene synthases (TPS) catalyse the terminal steps in the pathways and use geranyl diphosphate (GPP), trans-trans-farnesyl diphosphate (trans-trans-FPP, usually referred to as FPP) and geranylgeranyl diphosphate (GGPP) to form monoterpenes (C10), sesquiterpenes (C15) and diterpenes (C20), respectively. There are many different TPSs, and differences in their catalytic mechanisms generate a wide range of different terpenoids [92,93]. TPS families have been identified in many plants, including Arabidopsis, tomato, grapevine, apple, orange and kiwifruit [94,95,96,97]. Monoterpenes and sesquiterpenes are volatile and contribute to the characteristic aroma of fruits, including citrus [98]. CitAP2.10 [99] and CitERF71 [100] have been reported to regulate the synthesis of the sesquiterpenes valencene and E-geraniol respectively, in citrus. Nieuwenhuizen et al. [101] showed that mutation in the NAC binding region of *TPS* promoters in two kiwifruit species influences their monoterpene contents.

## 3. Ripening Is Influenced by Multiple Hormones and Light

The initiation and coordination of fruit development and ripening is complex and still not completely understood. Ethylene production is initiated at the onset of ripening and mature fruits have the ability to both synthesise their own ethylene and to respond to external ethylene by initiating ripening [4]. There is considerable evidence that hormones other than ethylene are also involved in fruit development and ripening (discussed in [102]). Auxin has been known for many decades to be required for fruit growth in strawberry [103], and gibberellins and auxins cause parthenocarpic fruit development in tomato [104]. Auxins and cytokinins can circumvent the requirement for fertilisation and stimulate development of parthenocarpic fruits [105]. It is possible that signals such as abscisic acid (ABA) from mature seeds could trigger the onset of ripening, but the fact that parthenocarpic tomatoes, bananas and grapes ripen without seeds indicates that such signalling is not an absolute requirement for ripening to occur [106]. There are now many reports showing that auxin, ABA and jasmonic acid (JA) also influence expression of genes involved in the biosynthesis of ethylene and other aspects of the ripening control network, as discussed later.

### 3.1. Ethylene Initiates and Promotes Ripening

Ethylene production occurs at a low level throughout plant growth and development but increases during biotic and abiotic stresses, abscission, ripening and senescence. During ripening onset and progression, there is a massive burst of ethylene production, which occurs in two stages, called system-1 and system-2 ethylene synthesis [12,107,108,109], and when system-2 is activated, ethylene synthesis becomes autocatalytic. In tomato, 14 *ACC synthase* (*ACS*) genes (*ACS1A/B-13*) and 7 *ACC oxidase* (*ACO*) genes (*ACO1-7*) have been identified [110,111]. *ACS1A* and *ACS6* are mainly involved in system-1 ethylene production, and *ACS2* and *ACS4* in system-2 [110,111]. *ACO1* participates in system-1 ethylene synthesis and *ACO1* and *ACO4* are involved in system-2 ethylene in tomato, but the *ACO* gene family has not been as well-studied as the *ACS* family [12,110]. *ACO1* expression increases strongly and correlates well with the autocatalytic rise in ethylene production. *ACO2* expression drops to a basal level during further fruit development and ripening. At the onset of ripening, *ACO3* expression strongly increases. There is a temporal increase in *ACO4* expression during the breaker stage, mainly in the columella and placenta tissue. *ACO5* expression increases slightly after anthesis and remains at a similar level during further fruit development and ripening. *ACO6* has a low expression during fruit development but a temporary high expression during the breaker stage, followed by a gradual decline during further ripening. *ACO7* is only basally expressed during fruit development and ripening [110]. Antisense gene silencing was used to test the function of *ACS* and *ACO* genes expressed during tomato ripening, and this resulted in the inhibition of ethylene synthesis and slowed or prevented both fruit ripening and leaf senescence [107,108,109], which validated the conclusions drawn about the role of ethylene from experiments with Ag+ (plant ethylene responses inhibitor [112]) and confirmed using 1-MCP (a specific inhibitor of ethylene perception and action [113]). Theologis [114] also showed that the respiratory climacteric was abolished in tomatoes in which *ACS* genes were inhibited. They did not produce ethylene, but ripening was restored by supplying ethylene externally. The roles of ethylene biosynthesis genes have been functionally verified in other fruits as well, such as melon [115], pear [116], kiwifruit [117] and apple [118].

Several TFs have been shown to influence ethylene synthesis, including MADS-RIN [7,119], HB-1 [120], NAC1/4 [121,122] and NAC4/9 [123,124] in tomato, and also apple MdERF2 [125], kiwifruit AdNAC6/7 [20,126] and banana MaERF9/11 [127]. The relationship between MADS-RIN and ethylene in controlling ripening has recently been clarified Figure 4 and Figure 5). Treatment of RIN-deficient fruit (engineered using CRISPR/Cas9) with propylene (an ethylene analogue used to study system-1 and system-2 ethylene [128]) showed that they were unable to initiate system-2 ethylene production [15]. RIN-deficient fruit undergo partial ripening, however, and if ethylene is added externally, they ripen slightly more, but not fully, whereas if 1-MCP is added just before the onset of ripening, ripening initiation and progression are almost completely inhibited for at least 20 days. This demonstrates that ethylene is sufficient to initiate ripening (Figure 4). This fits with the previous demonstration that RIN activates transcription of *ACS2* and *ACS4* [129], for system-2 ethylene synthesis. Li et al. [15] proposed a model (Figure 6) in which ethylene signalling initiates ripening, where RIN is required to upregulate system-2 ethylene synthesis (see Section 6) and also many other ripening-related genes are required for the progression of full ripening. Several other plant hormones (discussed later in Section 3) also upregulate specific *ACS* and *ACO* genes and stimulate ripening by promoting ethylene production.

At the genetic level, ethylene effects are brought about by ERFs. Tomato has 77 ERFs, which can be divided into nine subclades (A–J), and some are activators, while others are repressors. Members of the ERF F subclade possess the transcriptional repression EAR-motif [130]. A number of different ERFs have been shown to regulate individual genes or processes, contributing to changes in colour, flavour, texture and aroma in different fruits [131]. During ripening, transcripts for 27 ERFs accumulate, while mRNA levels for another 28 decrease [132], which suggests that different *ERFs* have contrasting roles in fruit development and ripening. Differences between *ERF* expression in the tomato ripening mutants *ripening inhibitor* (*rin*), *non-ripening* (*nor*), *Never-ripe* (*Nr*) and wild-type (WT) identified ERFs that were both strongly up- and down-regulated during normal ripening. Three *ERFs*, members of sub-class E, were dramatically downregulated in the mutants, indicating that they probably had important roles in controlling ripening events [132]. Several tomato *ERFs* have been shown to be involved in fruit softening, probably by mediating ethylene production [133], and *ERFs* have been identified in ripening apple, banana, citrus, grape, kiwifruit, persimmon and tomato [56,134,135,136]. AP2a, which belongs to the AP2/ERF family, appears to be involved in repressing ethylene production, since reducing *AP2a* expression results in enhanced ethylene production and softer fruits [133]. One problem associated with the designation of TFs as ERFs is that the attribution is most often made on the basis of computer sequence analysis and homology rather than functional assay. This overlooks the possibility that some so-called ‘ERFs’ are actually regulated by hormones other than ethylene, such as auxin, ABA, JA, etc., and similar arguments can be advanced for other TF families, such as the auxin response factors (ARFs), discussed in Section 3.2. This could explain why some ERF-type genes are expressed in non-climacteric fruits, where hormones other than ethylene are important in ripening regulation. Future challenges will involve unravelling the molecular mechanisms underlying the specificity of ethylene responses during plant development and fruit ripening. Deciphering the function of *ERF* genes in both ethylene-dependent and ethylene-independent processes during ripening and identifying the target genes of individual ERFs will be instrumental in clarifying their specific contribution to fruit ripening [111].

### 3.2. Auxin Delays Ripening and Antagonises the Effects of Ethylene

Physiological studies suggested that auxin plays an inhibitory role in fruit ripening [137,138], and reverse genetics experiments support this idea [139,140,141,142,143]. Low internal auxin concentration or reduced auxin signalling activity are believed to increase the sensitivity of fruit tissue to ethylene, promoting the transition from system-1 to system-2 autocatalytic ethylene production.

Auxin can be synthesised in the chloroplasts from tryptophan and then via either indole-3-acetamide or indole-3-pyruvic acid, to generate IAA [144]. There is also an important tryptophan-independent auxin biosynthesis pathway in the cytosol, involving TAA and a multigene family of *YUCCA* genes, which encode flavin-containing monooxygenases [145,146]. There are three types of auxin transcriptional regulators, ARFs, Aux/IAA and TOPLESS (TPS) proteins. ARFs play a key role in regulating the expression of auxin response genes and act in concert with Aux/IAAs to control auxin-dependent transcriptional activity of target genes. There are 47 *ARF* genes in banana [147], 22 in tomato [148] and 19 in sweet orange [149]. Most have an N-terminal DNA-binding domain, a variable central transcriptional regulatory region, which can function as an activator or repressor domain, and a carboxy-terminal dimerization domain, that allows formation of either ARF/ARF homodimers or ARF/Aux/IAA hetero-dimers.

Experiments with several ARFs have indicated that they play a role in tomato fruit development and ripening. The over-expression of tomato *ARF2A* resulted in a blotchy ripening pattern, with some parts of the fruit ripening faster than others [140]. The downregulation of tomato *ARF4* also altered ripening-associated phenotypes, such as firmness, sugar and chlorophyll content, leading to dark green fruit and blotchy ripening [143,150], and Yuan et al. [151] found a similar phenotype with tomato *ARF10*, which is involved in regulating chlorophyll and sugar accumulation during tomato fruit development [151]. *ARF2A* expression is reduced in the *nor*, *rin* and *Nr* ripening mutants and responds to exogenous application of ethylene, auxin and ABA [140]. ARF2A homodimerizes and also interacts with the ABA STRESS RIPENING (ASR1) protein, suggesting that ASR1 could link ABA and ethylene-dependent ripening [140]. Both auxin and ethylene *cis*-regulatory elements are present in the promoter regions of a number of *ARFs*, suggesting that they may be regulated by both hormones [148,152]. Furthermore, AtARF7 and AtARF19 are believed to be involved in the ethylene response in Arabidopsis [153] and several *ERF* genes are also induced by auxin [154,155]. The two *ARF2* paralogs in the tomato genome, *SlARF2A* and *SlARF2B*, are nuclear localised where they repress auxin-responsive genes. In fruit tissues, *SlARF2A* is ethylene-regulated, while *SlARF2B* is auxin-induced [141]. Taken together, these results indicate substantial interactions between ethylene and auxin in the regulation of ARFs and ERFs. SAURs (small auxin-upregulated RNAs) are believed to be involved in various auxin-related actions [156], and *Sl-SAUR69*, which shows reduced expression in the *rin* mutant, is involved in changing auxin signalling or transport. Overexpression of *Sl-SAUR69* in tomato causes premature initiation of ripening, whereas its downregulation delays ripening initiation [139,142].

### 3.3. Abscisic Acid (ABA), Jasmonic Acid (JA) and Brassinosteroids (BR) Promote Ethylene Synthesis and Fruit Ripening, while Salicylic Acid (SA) Inhibits Ripening

#### 3.3.1. ABA

There are many reports of the effects of ABA on ripening, as discussed by Zhu et al. [157] and reviewed recently by Kou et al. [158,159,160]. ABA is synthesised from carotenoids via the action of 9-cis-epoxycarotenoid dehydrogenase (NCED) [161,162]. Treating fruits of tomato, apple, grape, kiwifruit, peach and strawberry with ABA or ABA inhibitors, either enhanced or retarded the production of carotenoids, anthocyanins, volatiles, pectolytic enzymes and softening [158,163,164,165,166,167].

ABA can increase the levels of ACS and ACO activity and their transcripts [168,169], which would be expected to promote various aspects of ripening by increasing ethylene production. Several TFs have already been identified that activate transcription of *ACS* and *ACO* genes, such as RIN, HB-1 and ERFs [111,120,129], and it is important to establish which TFs respond to ABA in order to upregulate ethylene biosynthesis genes. It has been shown that NAC19/48 in tomato can directly induce the expression of *ACO1* and *ACS2,* and virus-induced gene silencing (VIGS) of tomato *NAC4* and *NAC9* reduced *ACS2*, *ACS4* and *ACO1* expression. Interestingly, there are binding sites for ABF (ABA responsive element-binding factor) TFs in the promoter regions of *NOR* and *RIN* genes [170], so the question arises: does ABA promote ethylene synthesis and ripening through an effect on these genes, as well as having an effect on ethylene synthesis?

#### 3.3.2. Jasmonates

Jasmonates (jasmonic acid (JA) and methyl jasmonate (MeJA)) are important plant hormones derived from linolenic acid [171]. Work on several fruits has supported a role for JA in modulating ethylene synthesis and ripening, and in apple, JA induces ethylene synthesis by enhancing expression of *MdMYC2*, which is known to be involved in JA signalling. MdMYC2 promotes ethylene biosynthesis by binding to the promoters of *MdACO1* and *MdACS1* and by regulating apple ERFs [172]. However, if the expression of *MdACS1* was blocked by 1-MCP, the MeJA treatment was ineffective in enhancing ethylene production. More recently, Wu et al. [173] showed that external ethylene and MeJA increased ethylene production, and this was correlated with higher transcripts of *ACS* genes *AdACS1* and *AdACS2,* and ACS enzyme activity in kiwifruit.

#### 3.3.3. Brassinosteroids

The brassinosteroids (BRs) are steroid hormones that play key roles in plant development and defence, and they also have a strong influence on aspects of fruit ripening in several fruits, including tomato, strawberry and persimmon. Application of brassinolide (BL, the most active brassinosteroid) to tomato fruit enhanced the accumulation of transcripts encoding the ethylene biosynthesis genes *ACS2*, *ACS4*, *ACO1*, *ACO4* and the key carotenoid biosynthesis gene *PSY1*, and there was a concomitant acceleration of tomato fruit ripening [174]. Application of the BR biosynthesis inhibitor brassinazole (BZ) to persimmon fruit delayed aspects of ripening, and when the brassinosteroid 24-epibrassinolide (EBR) was applied externally, transcripts of *DkACO2*, *DkACS1* and *DkACS2,* plus several genes encoding cell wall-modifying enzymes (*DkPG1*, *DkPL1*, *DkPE2*, *DkEGase1*), were upregulated. Furthermore, application of BR to large green strawberries promoted ripening, whereas the inhibitor BZ inhibited ripening. Furthermore, downregulation of brassinosteroid receptor *FaBRI1* expression by VIGS also retarded the development of red colour in green strawberry fruit [175].

#### 3.3.4. Salicylic Acid (SA)

Plants synthesise salicylic acid (SA) from chorismite in the plastids, via either the isochorismate synthase (ICS) and/or the phenylalanine ammonia-lyase (PAL) pathways [171,176]. The volatile derivative methyl salicylate (MeSA) is generated by methylation of SA by a specific form of O-methyltransferase [177]. Application of SA delays ripening of banana and kiwifruit [178,179] and there are several reports of the retardation of ripening by MeSA in mangos, papayas and sweet peppers [180,181]. The effect of MeSA, however, may depend on dose and time of application. Ding and Wang [182] showed in tomato that low concentrations (0.1 mM) of MeSA applied at the mature green stage and 0.01 mM of MeSA at the breaker stage enhanced the production of colour, ethylene and respiration. At a higher concentration (0.5 mM), however, MeSA prevented ethylene production, respiration and colour development, and this was associated with suppression of accumulation of *ACS2* and *ACS4* transcripts and delayed accumulation of *ACO1*. MeSA also delays processes such as softening, in addition to ethylene synthesis and colour change, but it is not clear whether the delay in ripening is due exclusively to the inhibition of ethylene biosynthesis transcripts or whether other ripening genes are also directly affected.

### 3.4. Influence of Light of Different Wavelengths on Ripening

Plants have several photoreceptor proteins containing chromophores that sense different regions of the visible and UV spectrum, including the phytochromes (PHYs) that perceive red (R) and far-red light (FR) and measure the R/FR ratio, the cryptochromes (CRYs), phototropins and ‘Zeitlupes’ that sense blue/UV-A light and the UV-B receptor UVR8, which senses UV-B light via a cluster of tryptophan residues [183].

The characterisation of the *high-pigment* (*hp1* and *hp2*) tomato mutants contributed to understand the role of light signalling during plant development and ripening. Ripe fruits of these mutants have higher levels of ascorbic acid (vitamin C), carotenoids, flavonoids and tocopherol (vitamin E) [184,185,186], and there is a general stimulatory effect of light on isoprenoid metabolism in fruit and vegetative tissues [187,188]. These effects result from mutations in negative regulators of light signalling: *hp1* is mutated in the nuclear protein UV-DAMAGED DNA BINDING PROTEIN1 (DDB1) and *hp2* is mutated in a second nuclear protein, DEETIOLATED1 (DET1). The WT versions of these proteins are negative regulators of light signal transduction [185,189,190], and RNAi silencing of *Sl-DDB1/HP1* or *Sl-DET1/HP2* increased plastid biogenesis and carotenoid accumulation in tomato fruit [191,192]. Silencing of three other tomato light signalling regulators (CUL4, COP1, PIF1a) also increased fruit carotenoid levels [185,192,193], and suppressing the light signalling hub gene *HY5* produced an opposite phenotype. Recent work has shown that tomato *HY5* (ELONGATED HYPOCOTYL5) regulates fruit ripening by targeting genes involved in carotenoid biosynthesis and ethylene signalling, and may also affect the translation efficiency of a set of ripening-related genes by targeting ribosomal protein genes [194].

Cruz et al. [195] showed that the high carotenoid content in ripening *hp2* fruits was associated with disturbed ethylene production, increased ethylene sensitivity and altered expression of several ripening TFs, including the downregulation of *ERF.E4*, a repressor of carotenoid synthesis, and altered auxin signalling. This was accompanied by severe downregulation of *AUXIN/INDOLE-3-ACETIC ACID* (*Aux/IAA*) genes and altered accumulation of *ARFs* transcripts, with ARF2 proteins (ARF2a and ARF2b), which are involved in tomato fruit ripening.

Thus, it is clear that there are strong interactions between light signalling, auxin and ethylene in tomato fruit, and that each make significant regulatory contributions to carotenoid biosynthesis and tomato fruit ripening [195]. Further work is required, however, before we can fully understand the molecular basis for the interactions between light, auxin and ethylene in the control of ripening.

## 4. Accessibility of Genes for Transcription during Fruit Development and Ripening

Young fruits do not ripen in response to external ethylene until they reach a certain stage of maturity, and it is clear that a mechanism exists for inhibiting premature ripening. This restricts the expression of ripening genes until the appropriate stage of development. This has the advantage that predators are not attracted to eat the fruit until the seeds are mature and ready for dispersal. This restriction is not fully explained by the absence of the required TFs or the presence of gene repressors. Rather, the accessibility of genes during development is reversibly regulated by epigenetic modifications to chromatin that govern the availability of genes for transcription. This involves changes to DNA methylation status of specific cytosine residues and also modifications to histones, particularly methylation and acetylation on conserved lysine residues of histone H3, although other modifications may also be important.

### 4.1. DNA Methylation

DNA methylation is influenced by DNA methylases and demethylases, and, in general, hypermethylation of DNA is associated with gene repression. In 1991, differential DNA methylation was first reported in two tomato genera by Messeguer et al. [196]. Zhong et al. [197] confirmed that epigenome status was not static during fruit development, and Liu et al. [198] demonstrated an association between tomato fruit development and genomic DNA demethylation that was governed by DNA demethylase 2 (DML2). In addition, a transient increase in DNA methylation was observed during chilling of tomato fruits [83], which represses expression of flavour genes. DNA methylation also represses gene expression during fruit development in grape [199], strawberry [200] and orange [201].

Methylation occurs on the 5 position of the cytosines in CG, CHG and CHH contexts (where H = A, T or C), often in gene promoters, and is maintained by methyltransferase1 (MET1) and chromomethylases (CMT3 and CMT2) [202,203]. DML2 actively removes methyl groups from methylated DNA during fruit ripening [198], and genome-wide DNA methylome analysis of the *sldml2* mutant of tomato has shown a strong direct correlation between the demethylation of hundreds of ripening-associated genes and their expression [204], including important ripening regulators such as tomato SPL-CNR, MADS-RIN and NAC-NOR in tomato [197,198,200,201,204], which are discussed in detail in Section 6. Recently, the *gs* mutant form of *GREEN STRIPE* (*GS*) in tomato has been identified as a methylated isoform of *TAGL1*, an important fruit development and ripening TF, which regulates chloroplast development and carotenoid accumulation in tomato fruit [205]. In this mutant, the *TAGL1* gene is methylated and the mutant fruit have patches where ripening is repressed. High methylation of the *TAGL1* promoter is correlated with downregulation of its expression, which increases chlorophyll content and delayed appearance of PSY1 and carotenoid accumulation, leading to dark green stripes on ripening tomatoes, with light green stripes in regions where *TAGL1* is less methylated. This stripe pattern is retained and is converted to yellow and red stripes as the fruit ripen.

### 4.2. Histone Marks

Histone post-translational modifications (PTMs) are also involved in regulating the accessibility of genes for expression. PTMs include phosphorylation, methylation, acetylation or ubiquitination, and depend on a wide range of enzymes that determine their distribution and abundance [206]. Several different histone marks have been identified in plants by chromatin immunoprecipitation combined with microarray analysis (ChIP-chip) or deep sequencing (ChIP-seq), and histone modification, as recently reported [207]. Functional histone marks include H3K9ac (histone H3 lysine 9 acetylation), H3K4me3/2/1 (histone3 lysine K4 trimethylation/demethylation/monomethylation) and H3K27me3 (histone lysine 27 trimethylation). The removal or addition of these marks is associated with gene activation or repression and is, to some extent, conserved in animals and plants [204,208,209,210,211,212,213,214]. In rice, condensed heterochromatin, which often contains inactive genes, has less H3K4me2 and H3K4me3 and more methylated DNA than the less compact euchromatin [215], and genes with predominantly H3K4me3 are actively transcribed, whereas genes with predominantly H3K4me2 are transcribed at lower levels. H3K9ac and H3K4me3 have been correlated with gene activation during different developmental processes in plants [216,217,218,219,220,221,222]. Zhang et al. [223] showed that three types of H3K4 methylations were present in Arabidopsis (H3K4me1, H3K4me2 and H3K4me3), and H3K4me-containing genes were highly expressed, although only H3K4me3 was thought to be directly related to transcriptional activation.

Tomato proteins HP1 and HP2, involved in fruit carotenoids metabolism, which are affected in the tomato mutations *hp1* and *hp2*, are capable of forming a complex with CULLIN 4 (CUL4), a ubiquitin E3 ligase component that participates in chromatin remodelling [224]. Two tomato polycomb repressive complex 2 (PRC2) constituents, the ENHANCER OF ZESTE homologs SlEZ1 and SlEZ2, encoding putative histonelysine-*N*-methyltransferases, are involved in fruit morphology [225,226]. Liu et al. [132] showed that a ripening-related polycomb-group protein encoded by a *MULTICOPY SUPPRESSOR OF IRA1* (*MSI1*) gene, *SlMSI1*, inhibits ripening, possibly through repression of *MADS-RIN* expression.

In contrast to histone marks associated with gene activation, the trimethylation of lysine 27 at histone H3 (HK27me3) by polycomb complexes is a conserved gene silencing mechanism [227]. Polycomb repressive complexes 1 and 2 (PRC1 and PRC2) are involved in histone modifications that cause the epigenetic repression of animal and plant genes [228]. PRC1 is required for proper H3K27me3 deposition to specific targets [229], but the animal and plant versions have diverged [230]. Recently, Liang et al. [231] showed that in tomato, one component of PCR1 (Like Heterochromatin Protein 1 (LHP1)) has two variants, SlLHP1a and SlLHP1b. *SlLHP1b* is expressed during ripening and co-localizes with H3K27me3 marks in chromatin from tomato fruit. Inhibiting *SlLHP1b* by RNAi in tomato fruits upregulated genes for ethylene synthesis (*ACS2*, *ACS4*, *ACO1*) carotenoid production (*PSY1*) and cell wall modification (*PL*), and the regulator genes *MADS-RIN* and *NAC-NOR* were also upregulated, whereas overexpressing *SlLHP1b* in fruits downregulated these genes [231].

Li et al. [232] identified SlJMJ6 as a H3K27me3 demethylase that specifically demethylates H3K27me3, thereby activating the expression of ripening-related genes, and showed that overexpression of tomato *SlJMJ6* accelerated tomato fruit ripening. RNA-seq and chromatin immunoprecipitation identified 32 genes directly targeted by SlJMJ6, including ripening-related genes such as *MADS-RIN, ACS4*, *ACO1*, *PL*, *TBG4* and *DML2*.

Hu et al. [233] recently provided evidence that histone post-translational modifications, rather than DNA methylation, underlie changes in gene expression during pollination-dependent and hormone-induced fruit set in tomato. Differentially expressed genes were mainly associated with changes in H3K9ac or H3K4me3. Highly expressed genes showed extremely low association with H3K27me3 but were highly enriched in H3K9ac and/or H3K4me3, although it was found that some genes or chromatin regions may carry both types of marks [233]. The demonstration by Li et al. [232], however, that *SlJMJ6* overexpression upregulates *DML2*, in addition to other known ripening genes, establishes a direct link between histone H3K27 demethylation and DNA demethylation, and it seems that both DNA methylation and histone modification are involved in regulating expression of ripening genes.

Other changes to histones can also have a profound effect on ripening. For example, Yang et al. [234] used the CRSIPR/Cas9 gene-editing system to generate a tomato double mutant of the histone variant *Sl*_H2A.Z. This reduced the fresh weight of tomato fruits and increased the contents of carotenoids by increasing the expression of genes *SlPSY1*, *SlPDS* and *SlVDE*, which encode enzymes in the carotenoid biosynthesis pathway.

## 5. Non-Coding RNAs

Diverse types of noncoding RNAs have been reported, including microRNAs (miRNAs), small interfering RNAs (siRNA), trans-acting siRNAs (tasiRNA), long noncoding RNAs (lncRNAs), natural antisense transcripts (NATs) and circular RNAs (circRNAs), which play a variety of roles in an RNA network regulating gene expression during development and ripening.

MicroRNAs (miRNAs) are small non-coding RNAs, of typically 20–24 nt, which regulate gene expression either transcriptionally or post-transcriptionally through sequence complementarity with target genes and/or their mRNAs. Ripening-induced DNA hypomethylation is associated with decreased siRNA levels, consistent with reduced RdDM activity [200]. These 24 nt siRNAs are further methylated by HEN1 [235,236,237] and interact with scaffold RNAs and participate in the RNA-directed DNA methylation (RdDM) gene silencing pathway.

Recent progress in understanding the role of miRNAs in regulating fruit development and quality characteristics during ripening and senescence has been summarised by Ma et al. [238,239]. miR172a is inhibited by MADS-RIN transcription factor, and several miRNAs have been shown to modulate the ethylene signal transduction pathway (miR1917) [238] and ethylene biosynthetic genes *ACS8* (miR159) and different *ACO* genes (miR414, miR396b, miR397, miR1917) [239]. Recently, miR164 was found to cleave *AdNAC6/7* mRNAs and inhibit their transcriptional regulatory effects on ripening-related genes in kiwifruit [20].

CircRNAs can arise from exons (exonic circRNA), introns (intronic circRNA) and intergenic regions [240,241], and expression of some exonic circRNAs and their parent genes are significantly positively correlated [242]. Most plant circRNAs show developmental-/stress-specific expression patterns, as reported in animals [243], and are involved in regulating fruit colouration [244,245], fruit development [246], fruit chilling response [247] and the fruit ethylene biosynthetic pathway [248].

LncRNAs are broadly defined as non-coding RNAs longer than 200 nucleotides in length [249]. Recent work suggests that lncRNAs play critical roles in transcriptional and post-transcriptional regulation [250], as well as in epigenetic modification, cell differentiation and development [251], and they can recruit polycomb-repressive complexes, alter trans-interactions and chromatin function, activate or suppress gene transcription and expression, alter RNA splicing or mRNA stability or activate mRNA translation and protein post-translational modifications [252,253]. Several lncRNAs are reported to regulate tomato fruit development and ripening [254,255], although knowledge of the transcriptional regulation mechanism of lncRNAs is limited [256]. Multiple lncRNAs regulate the ethylene biosynthesis pathway by controlling the expression of different *ACS* and *ACO* targets, lncRNAZ018 targets *ACS2*, while lncRNAZ113 and lncRNAZ118 target *ACO2* [239]. The lncRNA2155 is involved in the regulation of tomato fruit colour, and repression of *lncRNA1459* inhibited lycopene accumulation [239]. Tomato lncRNA192 and lncRNA291 target *flavonol synthase* (*FLS*), while the targets of lncRNA135, lncRNA027 and lncRNA143 are *L-ascorbate peroxidase 3* (*APX3*), *lipoxygenase* (*LOX*) and *vitamin K epoxide reductase* (*VKOR*), which are related to the fruit flavour in tomato [239]. Additionally, the target gene of tomato lncRNAZ180 and lncRNA TCONS_00028129 is *pectinesterase* (*PE*), and the target gene of lncRNA TCONS_00007262 is *pectate lyase* (*PL*), which indicate that lncRNAs may regulate fruit texture change [239]. Further clarification of the diverse mechanisms of action of lncRNAs is required in order to fully understand their role in modulating ripening.

## 6. The Identification and Function of Major Regulator TFs

Developments in map-base cloning helped focus attention on four spontaneous ripening mutations: *rin*, *nor*, *cnr* and *Nr*, identified by tomato breeders, that produced pleiotropic non-ripening mutant phenotypes. Genetic analysis suggested that WT versions of these genes were absolutely required in order for ripening to occur, and they were considered as candidates for ‘master regulators’. The *Nr* mutation was shown to encode an altered ethylene receptor that functioned as a negative regulator that was unable to bind ethylene [257]. Map-based cloning showed that *rin* encoded a MADS TF, MADS-RIN, *nor* encoded a NAC TF, NAC-NOR, and *cnr* encoded a SQUAMOSA promoter-binding protein-like TF, SPL-CNR. In addition to these spontaneous mutants, other TFs have also been identified such as TOMATO AGAMOUS-LIKE1 (TAGL1), APETALA2a (AP2a) and FRUITFULL (FUL1 and FUL2) [4]. TAGL1 (also mentioned in Section 4.1) is highly expressed during fruit ripening, interacts with MADS-RIN [258] and is a candidate for controlling ripening processes. Silencing the encoding *FUL* genes separately resulted in very mild alterations in tomato fruit pigmentation, indicating that FUL1 and FUL2 probably have redundant functions and FUL1/2 and TAGL1 may regulate different subsets of the known MADS-RIN targets. Further, *MADS-RIN* and *TAGL1* were found to be upregulated in the pericarp of *FUL1/2* RNAi fruits, pointing to a negative feedback loop from FUL1/2 to these genes [4].

Recent experiments using CRISPR/Cas9 technology [259] to generate mutants deficient in MADS-RIN, NAC-NOR and SPL-CNR produced a much less severe ripening phenotype than that shown by the original spontaneous mutants *rin*, *nor* and *Cnr*, and this has led to a complete reappraisal of their role in ripening [14,15,16,17,18,19,260].

### 6.1. Cnr (SPL-CNR) Methylation and Function

Plants carrying the *Cnr* mutation appear normal, but the ripe fruits have a yellow skin with colourless pericarp and show a loss of cell-to-cell adhesion, which affects the fruit texture. Map-based cloning and sequencing identified CNR as an SBP-box protein (SPL-CNR) [261], a TF with two zinc-finger motifs in the C-terminal SBP-box domain. The *Cnr* mutation does not affect the *SPL-CNR* coding sequence itself, but a 286 bp DNA region of the gene promoter, approximately 2.4 kb upstream of the coding sequence, is hypermethylated in the mutant. VIGS of *SPL-CNR* in WT tomato, using a Potato Virus X-based vector, led to development of non-ripening fruit sectors.

Further analysis showed that the *Cnr* epigenome is hypermethylated [197,262]. These additional methylation marks are not normally found in WT, which might be due to lack of expression of tomato *DEMETER-Like DNA demethylase 2* (*DML2*) [198]. Kanazawa et al. [263] used a Cucumber Mosaic Virus-based vector containing double-stranded RNA homologous to the *Cnr* promoter to induce promoter-silencing in WT plants. This caused hypermethylation of the targeted *Cnr* promoter and partially suppressed ripening. Inhibiting the accumulation of tomato *DML2* by RNAi also inhibited ripening by causing DNA hypermethylation, which repressed the genes encoding TFs and their downstream gene targets required for ripening [198]. Chen et al. [262] showed, using VIGS, that CHROMOMETHYLASE3 (SlCMT3) and other methyltransferases are required for maintenance of the *Cnr* phenotype.

These results pointed to the conclusion that SPL-CNR plays an important role in ripening control and that the *Cnr* mutation affects epigenomic methylation that inhibits normal expression of SPL-CNR. When CRISPR/Cas9 was used to generate a SPL-CNR deletion mutant of tomato, however, surprisingly, the fruit ripened more fully. RNA-seq analysis of the *Cnr* mutant and *CNR*-deficient fruits showed that the expression of ripening genes such as *ACS2*, *ACO1*, *PSY1*, *PG* and *EXP* are partially expressed in the CRISPR/Cas9 lines but repressed in the natural mutants [262]. These experiments raised serious doubts about the importance of *Cnr* in the control of tomato ripening [14,16,264,265]. The question remains how to reconcile the differences in phenotypes between the analysis of the CRISPR and natural *Cnr* mutants. It is known that the *Cnr* mutation is located in a 13 kb DNA region in the tomato genome [239].The mutant phenotype must be due to a change in this 13 kb region, possibly caused by new methylation mark(s), and it is possible, therefore, that DNA hypermethylation in nearby regions might contribute to the *Cnr* non-ripening phenotype [262].

It seems that CNR plays a less critical role in ripening than previously thought, and its role needs further investigation. It seems possible that the *Cnr* mutation is a gain of function caused by some methylation change. If so, understanding the phenotype is best addressed by investigations on the mutant itself, rather than focusing on the WT alone. It is possible that other yet to be discovered structural or epigenetic changes in this 13 kb region may provide an explanation for the discrepancies between the phenotype of the naturally occurring mutant and the CRISPR/Cas9 *Cnr* mutations.

### 6.2. The Rin Mutation (RIN-MC)

The tomato *rin* mutation causes a severely inhibited ripening phenotype, involving the loss of the respiratory climacteric and the major ripening-associated increase in ethylene synthesis, and a severe reduction in pigment accumulation, flavour production and softening [266]. Hundreds of genes involved in many aspects of ripening-related pathways, such as ethylene synthesis (*ACS2* and *ACS4*) [129], cell wall modification (*PG*, *TBG4* and *EXP1*) [129] and volatile production (*LoxC*) [267] are direct RIN targets, and transcriptome, proteome and metabolome analyses confirmed the conclusion that RIN is a global tomato fruit-ripening regulator [268]. Recent research, however, has led to a re-evaluation of this conclusion.

The *rin* mutation was known to involve a deletion and fusion of parts of two adjacent *MADS* genes (*RIN* and *MC*) located on tomato chromosome 5. The resulting *RIN-MC* gene fusion was originally believed to generate a loss of function mutation [266] (Figure 5). However, Ito et al. [18] and Li et al. [15,17] showed that there were major discrepancies between the severe ripening-inhibited phenotypes of the *rin* mutant fruit and the phenotype of RIN-deficient fruit, which were able to undergo partial ripening. The *RIN-MC* fusion transcript was further shown to accumulate in significant quantities in *rin* mutant fruit and demonstrated to be translated into an active nuclear-localised TF, which has a strong repressor function [17]. It was concluded that *rin* was actually a gain of function mutation, cause by part of the *MC* sequence in the *RIN-MC* fusion that conferred the ability to repress the expression of some ripening genes. It is this repressor activity that severely inhibits the change in colour, flavour, texture, aroma and ethylene synthesis required for normal ripening to occur (Figure 5). This was confirmed by comparative transcriptome analysis of *rin* and *rin*-*35S*::*RIN-MC* RNAi, and AC and AC-*35S*::*RIN* RNAi fruits [269]. Greatly reducing *RIN-MC* transcripts by RNAi, deleting *RIN-MC* from the *rin* mutant, or deleting *RIN* from WT plants using CRISPR/Cas9 gene editing generated much less severe ripening phenotypes than *rin* [15,17,18]. Another important difference between *rin* fruit (expressing *RIN-MC*) and RIN-deficient fruit (obtained by CRISPR/Cas9) is that the latter soften extensively. Transcripts of some softening enzyme genes including *XTH5* and *XTH8* were expressed at higher levels compared to WT, which might explain why RIN-deficient fruits softened similarly to WT fruits at early ripening stages. At later stages, *MAN1*, *Mside7*, *MAN4a*, *TBG4*, *PG* and *PME2.1* transcripts were significantly more abundant than in WT, and the fruit softened even more than WT, which may indicate that, when present, RIN can actually repress their expression, but this needs experimental confirmation [15].

**Figure 5 cells-10-01136-f005:**
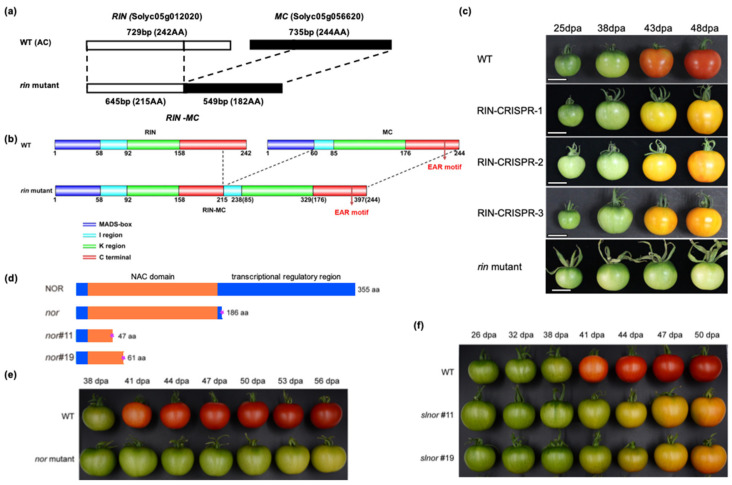
Re-evaluation of the roles of MADS-RIN and NAC-NOR. The structure of *rin* and *nor* mutant genes and their mutant phenotypes and differences compared to the corresponding CRISPR/Cas9 MADS-RIN and NAC-NOR mutants are shown. Redrawn from Li et al. [269] and Gao et al. [14]. (**a**) The *rin* mutation is the result of a partial deletion and fusion of 2 adjacent genes, *RIN* and *MC*. (**b**) Translated amino acids predicted for RIN and RIN-MC proteins. The colour key shows MADS-box, protein binding, I and K-region for protein structure formation, C-terminal for activation. EAR-motif indicates the repressor region. The numbers refer to amino acid positions and the colours indicate different protein domains. (**c**) Comparison of the phenotypes of the *rin* mutant and the RIN-CRISPR mutant. The ripening phenotype is much more severely inhibited in *rin* fruit compared to RIN-CRISPR [15,18] tomatoes. (**d**) Diagram of *nor* tomato mutation and NOR-CRISPR mutant (nor#11 and nor#19). WT tomatoes encode a full-length NAC-NOR protein of 355 aa. The *nor* mutant produces a truncated protein of 186 aa (NOR186). Comparison of the phenotype of the *nor* mutation (**e**) and the NOR-CRISPR mutant (**f**). The *nor* mutation produces a truncated protein (Nor186) which has an intact NAC domain. Nor186 causes a much more severe inhibition of ripening compared to the NOR-CRISPR mutant (nor#11 and nor#19), which lacks much of the NAC domain (from Gao et al. [14]).

Further experiments with these RIN-deficient fruits, together with other information concerning ripening behaviour discussed in Section 5, indicated that ethylene is necessary and sufficient to initiate tomato fruit ripening and RIN is not required for this. RIN is, however, absolutely required for full progression of ripening [15,17,18]. This is perhaps not surprising, given that WT RIN has been shown to have hundreds of gene targets and RIN-deficient fruits have greatly reduced levels of transcripts for some carotenoid biosynthesis and volatile biosynthesis enzymes, and low levels of genes required for system-2 ethylene production [15].

MADS proteins frequently form homo- or hetero-dimers and RIN has been shown to interact with FUL1/ FUL2, TAGL1 and SlMBP18, and can also act as a SEP-like protein bridging factor, generating higher-order protein complexes with several TFs, including TM4 and SlMBP24 [17], and if tomato FUL1/2 are inhibited, this also interferes with ripening [270]. *RIN-like* genes have also been shown to be involved in ripening of other climacteric and non-climacteric fruits [17]. There is an urgent need to investigate other *MADS* genes and potential RIN partners in a range of fruits in order to understand more fully how their interactions contribute to the ripening process.

### 6.3. The Nor (NAC-NOR) Mutation

The tomato *nor* mutation, like *rin*, produces a severe ripening-impaired phenotype (Figure 5). DNA sequencing showed that the mutation was due to a 2 bp deletion in the third exon of *NAC-NOR* [8]. This caused a frameshift, generating a transcript which is translated to produce a truncated protein of 186 amino acids (NOR186), which retains the NAC domain but lacks the C-terminal 169 amino acids (Figure 5). NAC-NOR-CRISPR gene editing experiments to generate NAC-NOR-deficient tomato plants produced a less severe phenotype [14,19], however, as also found for MADS-RIN (Figure 5). Wang et al. [19] suggested that the truncated NAC-NOR protein could cause a dominant-negative mutation, where the mutant TF could bind the DNA binding sites in target gene promoters and possibly also interact with other proteins but would be unable to transcriptionally activate its target genes. Gao et al. [14] showed that NOR186 is located in the nucleus and binds to, but does not activate, the promoters of *ACS2*, *GGPPS2* and *PL*, which are involved in ethylene biosynthesis, carotenoid accumulation and fruit cell wall softening, respectively. Furthermore, the activation of the promoters by WT NOR protein could be inhibited by the mutant NOR186 protein. In NAC-NOR-deficient fruit, ethylene synthesis, carotenoid accumulation and fruit softening were significantly inhibited compared with the WT, but much less severely affected than in the *nor* mutant.

### 6.4. The Wider Importance of the NAC Gene Family

In recent years, many lines of evidence have highlighted important roles for a range of NAC TFs in different aspects of plant development, including ripening. There are over 90 *NAC* genes in tomato and they often form homo- or hetero-dimers. Several, such as *SlNAC1* and *SlNAC4*, have been shown to affect ripening, in addition to NAC-NOR, and Gao et al. [271] carried out a systematic search of *NAC* genes expressed in fruit. Silencing *NOR-like1* (Solyc07g063420) using VIGS delayed ripening initiation by 14 days. When *NOR-like1* function was inactivated by CRISPR/Cas9, the tomato fruits produced less ethylene, showed retarded softening and chlorophyll loss and reduced lycopene accumulation. Gene promoter analysis suggested that NOR-like1 targeted genes involved in ethylene biosynthesis (*ACS2*, *ACS4*), colour formation (*GGPPS2*, *SGR1*) and cell wall metabolism (*PG2a*, *PL*, *CEL2* and *EXP1*), and this was confirmed by electrophoretic mobility shift assays (EMSA), chromatin immunoprecipitation-quantitative PCR (ChIP-qPCR) and dual-luciferase reporter assays [271].

*NAC1* is involved in tomato ripening, and manipulating its expression affects *ACS2*, *ACS4, ACO1* and ethylene biosynthesis, although there are conflicting reports about whether *NAC1* delays or accelerates ripening [121,124]. *NAC4* and *NAC9* have also been identified as positive regulators of tomato *ACS2*, *ACS4* and *ACO1* expression, and silencing of *NAC4* and *NAC9* repressed tomato fruit ripening [122,123] (see also Section 3.3.1 for the role of NACs in ripening regulation by ABA). NACs also play important roles in development and ripening of many other fruits, including apple, banana, kiwifruit, oil palm and peach [71,122,271,272]. Nieuwenhuizen et al. [101] showed that in kiwifruit, *AaNAC2/3/4* physically interacts with the promoter of the terpene synthase gene *AaTPS1*, which is important for monoterpene production. *AdNAC6/7* and *AdNAC2/3* in kiwifruit were also found to transcriptionally activate *AdACO1*, *AdACS1*, *AdMAN1* and *AdTPS1* [20]. In peach, the NAC transcription factor BL interacts with NAC1 to amplify its regulatory effect on anthocyanin biosynthesis, and in banana, NAC5 can interact with WRKY1/2 to enhance the induction of resistance genes [273,274]. MaNACs also can activate the promoters of *ethylene-insensitive* (*EIL*) genes during banana ripening [275]. These findings support a conserved regulation of genes involved in ethylene biosynthesis and other aspects of ripening by NAC TFs in different fruits, which, in addition, may also involve MYCs and ERFs.

## 7. Conclusions and Models for the Control of Ripening

At the molecular level, there are variations in ripening regulatory circuits that operate in different fruits, although many of the genetic components appear to be similar. Recently, Lü et al. [2] and Gao et al. [16] proposed that there are three types of circuits controlling ethylene synthesis and climacteric fleshy fruit ripening that utilise either MADS, NACs, or both types of TFs together. Each model involves the ethylene transcription factor EIN3. It is proposed that in tomato, EIN3 activates the MADS transcription factor RIN, which forms a complex with TAGL, and activates ethylene biosynthesis genes, forming a positive feedback circuit that generates autocatalytic ethylene during ripening, with RIN also regulating downstream ripening genes. In peach, ripening regulation utilizes a NAC instead of a MADS transcription factor, and in banana, an additional loop involving both *NAC* and *MADS* genes enables the banana fruit to synthesise ethylene in the presence of ethylene inhibitor 1-MCP after ripening initiation [2].

Several important conclusions can be drawn from studies on the roles of hormones and TFs in regulating fruit ripening:Addition and removal of repressive and promotive histone marks and DNA methylation are associated with the activation or repression of ripening genes and are an integral part of fruit maturation and preparation for ripening initiation and progression.Ethylene is responsible for the initiation of ripening in tomato and other climacteric fruits. Several other plant hormones, including ABA, JA and BR, promote ripening by upregulating ethylene biosynthesis genes. It is not yet clear whether these hormones also affect expression of other ripening genes.Ethylene biosynthesis and signalling genes and homologs of *MADS* and *NAC* genes are expressed in ripening of both climacteric and non-climacteric fruits.In tomato, auxin content and signalling decline during fruit maturation and auxin and SA tend to oppose the action of ethylene and inhibit ripening.MADS-RIN is required for the upregulation of many ripening genes, and progression and full ripening do not occur without MADS-RIN.*NAC-NOR* and several other *NAC* genes promote the expression of ripening genes involved in production of ethylene and change in colour, flavour and texture. Nor-like1 is important for the full expression of genes involved in ethylene biosynthesis (*ACS2*, *ACS4*), colour formation (*GGPPS2*, *SGR1*) and cell wall metabolism (*PG2a*, *PL*, *CEL2* and *EXP1*).Ethylene, acting through ERFs, enhances accumulation of tomato mRNAs for *MADS-RIN* and *NAC-NOR*. Conversely, NAC-NOR and Nor-like1 enhance ethylene biosynthesis by modulating the expression of specific *ACS* and *ACO* genes.Light, detected by different photoreceptors and operating through HY5, can also modulate ripening behaviour.

Figure 6 outlines the main TFs likely to interact with and modulate transcription from the promoters of genes encoding enzymes involved in fruit softening. Similar diagrams can be draw for promoter/TF interactions for a range of other ripening genes. A model outlining the main hormonal and genetic regulatory interactions governing the ripening of tomato fruit is shown in Figure 7. This model takes into consideration the role of histone marks, ethylene and other hormones which modulate ethylene biosynthesis genes, and MADS-RIN, NAC-NOR and other TFs that have been shown to be involved in the regulatory circuits. It should not be considered as a universal model for all fruits, but it serves to illustrate the complexity of ripening control mechanisms and may focus ideas to guide future experiments.

**Figure 6 cells-10-01136-f006:**
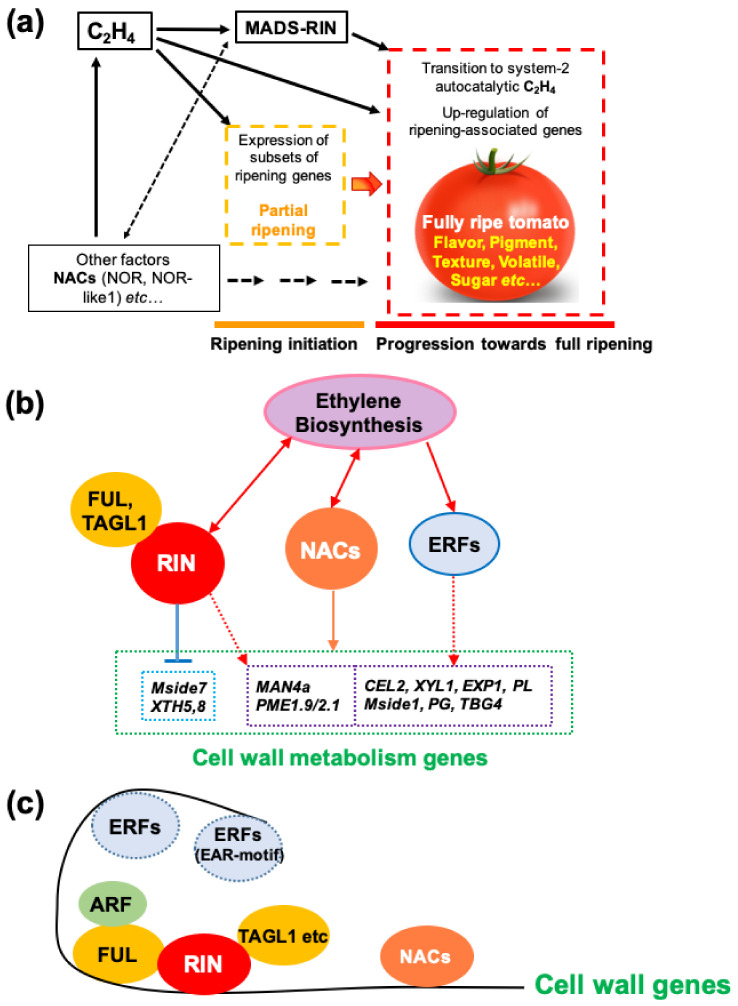
The range of TFs interacting with the promoters of genes encoding fruit ripening enzymes. Redrawn and modified from Li et al. [15]. (**a**) Model outlining the role of ethylene and RIN in initiation and progression of climacteric ripening in tomato fruit. Ethylene can initiate ripening in a RIN-independent way leading to partial ripening. However, RIN is required for autocatalytic system-2 of ethylene production and subsequent full ripening. *RIN* expression is enhanced by ethylene [2,73,269]. Other factors such as NACs (NOR, NOR-like1) affect ethylene production and are also involved in the ripening genetic program [14,15,271]. We only show the involvement of ethylene, but, as discussed in the text and in Figure 7, other hormones directly regulate some *ACS* and *ACO* ethylene biosynthesis genes, as does the transcription factor RIN. (**b**) A model of the role of RIN and ethylene in regulating tomato fruit cell wall changes and softening. Accumulation of *CEL2*, *XYL1*, *EXP1*, *PL*, *Mside1*, *PG* and *TBG4* transcripts is stimulated by ethylene [15]. RIN directly or indirectly regulates the transcription of genes involved in cell wall metabolism, such as *CEL2, XYL1, EXP1, PL, Mside1* and *PME1.9*. RIN also inhibits expression of genes such as *XTH5* and *XTH8*, but so far, there is no evidence that NACs including NOR and NOR-like1 inhibit transcripts of any of the cell wall metabolism genes. (**c**) Ethylene treatment affects expression of cell wall genes and their promoters contain ERF binding sites, which indicates that promoters of these genes might be activated by ERF TFs. RIN also targets cell wall genes and MADS proteins act in the form of protein complex. NACs also regulate softening genes, as indicated in (**c**). Tomato protein orthologues of FUL and ARF8 can also heterodimerise in vivo, suggesting that MADS-ARF associations may occur in diverse plant species [276]. ncRNAs appear to be involved in regulating ripening gene expression but they are omitted from the model until their diverse modes of action (Section 5) can be clarified. Subfigures (**a**,**b**) are modified from Li et al. [15].

**Figure 7 cells-10-01136-f007:**
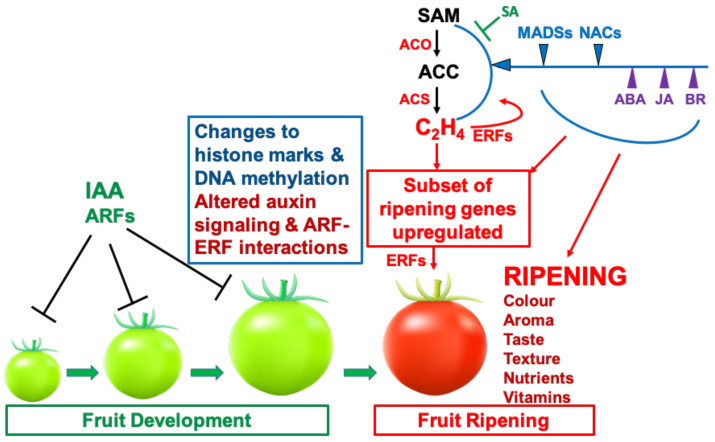
Model of tomato fruit ripening regulation. ERFs, ethylene response factors; ARFs, auxin response factors; SA, salicylic acid; ABA, ABA response element binding factors; JA, MYC Family bHLH TFs, etc.; BR, brassinosteroid response factors; MADSs include MADS-box TFs such as RIN, FUL1/2, TAGL1, etc.; NACs include NAC family TFs such as NOR, NOR-like1, etc.

## Figures and Tables

**Figure 1 cells-10-01136-f001:**
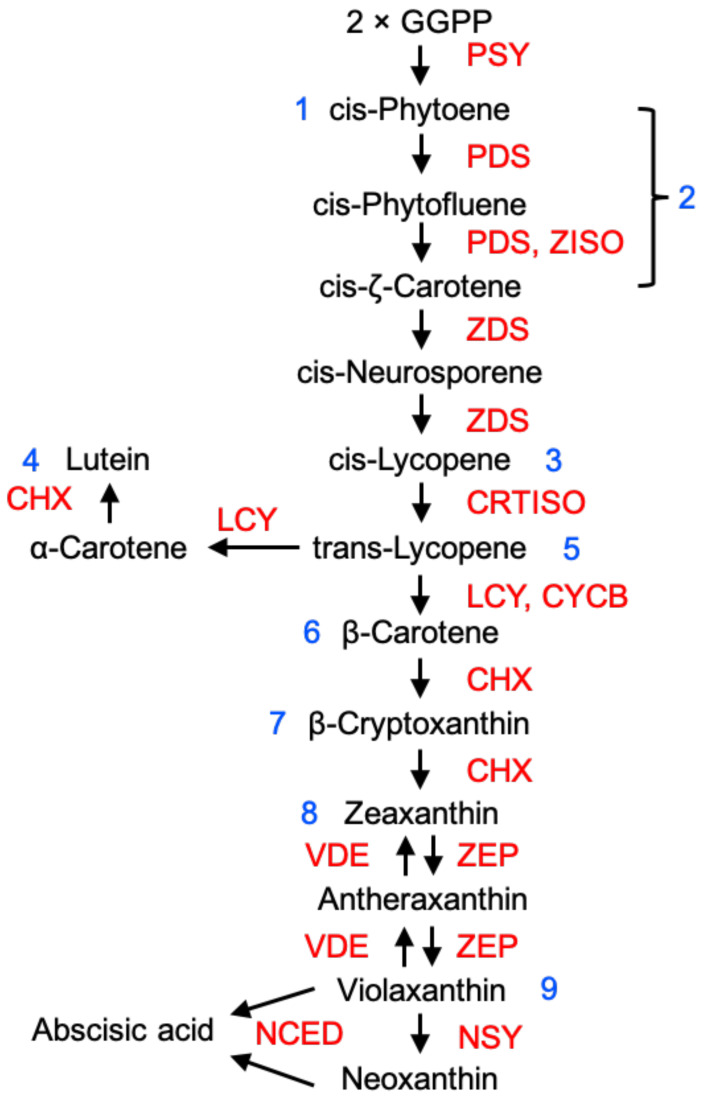
The carotenoid biosynthesis pathway and variations in different fruits (redrawn and modified from Lado et al. [33] and Luan et al. [30]). Each fruit labelled with a number in blue is positioned in the pathway at the level of the predominant carotenoid responsible for its coloration: **1**, Lemon (*Citrus limon*); **2**, Sweet orange *Pinalate* mutant; **3**, Tomato *tangerine* mutant; **4**, Grape (*Vitis vinifera*); Immature green pepper (*Capsicum annuum*); Kiwifruit (*Actinidia chinensis*); Peel of immature mandarin (*Citrus reticulata*, *Citrus*
*clementina*, *Citrus unshiu*); Avocado (*Persea americana*); **5**, Tomato (*Solanum lycopersicum*); Red watermelon (*Citrullus lanatus*); Pulp of red grapefruit (*Citrus paradisi*); Red papaya (*Carica papaya*); Gac (*Momordica cochinchinensis*); **6**, Orange-flesh melon (*Cucumis*
*melo*); Orange-flesh apricot (*Prunus armeniaca*); Orange-flesh pumpkin (*Cucurbita maxima*); **7**, Yellow papaya (*Carica papaya*); Loquat (*Eriobotyra japonica*); Pulp of mandarin (*Citrus reticulata*, *Citrus*
*clementina*, *Citrus unshiu*); **8**, Yellow-flesh peach (*Prunus persica*); Orange pepper (*Capsicum annuum*); **9**, Mango. PSY, phytoene synthase; PDS, phytoene desaturase; ZISO, ζ-carotene isomerase; ZDS, ζ-carotene desaturase; CRTISO, lycopene isomerase; LCY, lycopene cyclase; CYCB, chromoplast specific lycopene β-cyclase; CHX, carotene hydroxylase; ZEP, zeaxanthin epoxidase; VDE, violaxanthin de-epoxidase; NSY, neoxanthin synthase; NCED, 9-cis-epoxycarotenoid dioxygenase. Note that the hormone ABA, which is discussed in Section 3.3, is produced from this pathway. Some of these metabolites are also utilised in the biosynthesis of hormones, such as GA, strigolactones and β-cyclocital, but these pathways are omitted from this figure because they are not discussed in this review.

**Figure 2 cells-10-01136-f002:**
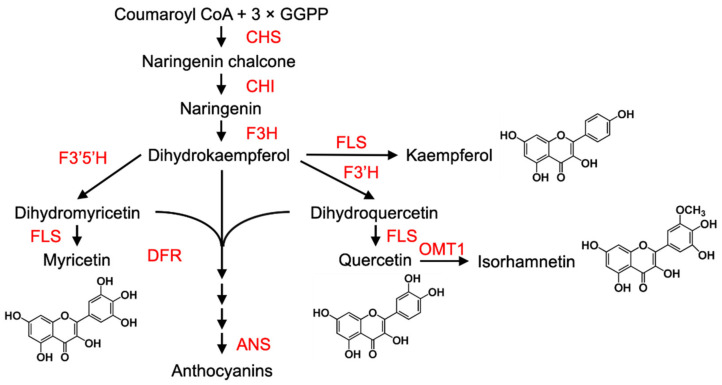
The anthocyanins biosynthetic pathway and variations in different fruits. This figure is redrawn and modified from Gayomba et al. [44]. The flavonol biosynthetic pathway is illustrated, showing enzymatic steps, and the responsible enzyme is indicated in red. CHI, chalcone isomerase; CHS, chalcone synthase; F3H, flavanone 3-hydroxylase; F3′H, flavonoid 3′-hydroxylase; F3′5′H, flavonoid 3′,5′-hydroxylase; FLS, flavonol synthase; DFR, dihydroflavonol reductase; OMT1, flavone 3′-*O*-methyltransferase 1.

**Figure 3 cells-10-01136-f003:**
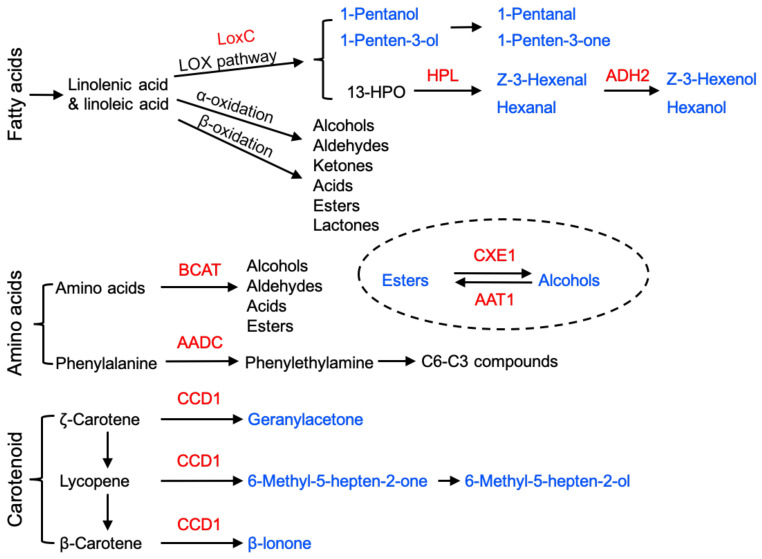
Biochemical origin of volatile compounds in fruits (redrawn from Aragüez and Valpuesta [86]; Klee and Tieman [88]). Key aspects of primary metabolism, which provides the substrates for secondary metabolism, are shown on the left. Solid lines indicate a validated step in a pathway, with the responsible enzyme indicated in red. Volatiles are indicated in blue. Broadly, volatiles are derived from the fatty acid (for example, Z-3-hexenol), carotenoid cleavage (for example, geranylacetone) or phenylpropanoid (the C6–C3 compounds) pathways. In addition, volatile alcohols can be reversibly converted to esters by the action of an alcohol acetyltransferase (AAT1) and a carboxymethylesterase (carboxylesterase 1 (CXE1)). LoxC, lipoxygenase C; HPL, fatty acid hydroperoxide lyase; ADH2, alcohol dehydrogenase 2; BCAT, branched-chain amino acid aminotransferases; AADC, aromatic amino acid decarboxylase; CCD1, carotenoid cleavage deoxygenase 1.

**Figure 4 cells-10-01136-f004:**
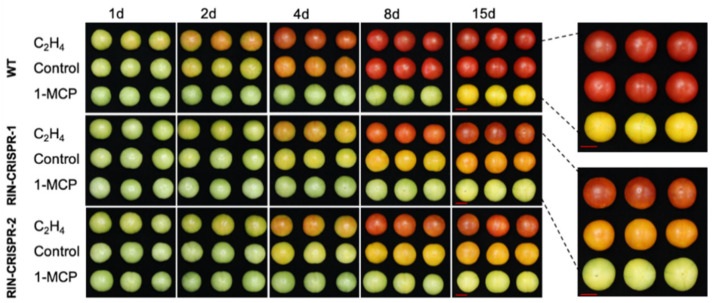
Ethylene response of WT and RIN-deficient tomato fruits (reproduced with the permission from Li et al. [15]). Effect of exogenous ethylene and ethylene perception inhibitor 1-MCP treatment on ripening progression of RIN-CRISPR tomato fruit. WT and RIN-CRISPR tomato fruits were picked at MG stages and treated and replenished daily with ethylene (100 ppm) and 1-MCP (10 ppm) or air continually for up to 15 days. Fruits in horizontal rows are biological replicates. Enlarged photos of representative samples are shown compared to WT fruits on the right. The red scale bar represents 2 cm.

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
