# Peer review of "Molecular and Hormonal Mechanisms Regulating Fleshy Fruit Ripening"

_cells, 2021, doi:10.3390/cells10051136_

Round 1

Reviewer 1 Report

The manuscript reviews the latest research on fruit ripening in fleshy fruit. I found it comprehensive and informative, although information dense at times. A number of aspects of the physiology of fruit ripening have been considered, from metabolic changes associated with fruit composition to transcriptional regulation. The title could be more informative, e.g. Molecular and hormonal mechanisms controlling ripening in fleshy fruit.

The authors should revise section 2.3, focusing on the sugar and starch metabolism. References are not accurate, and more details should be added.

Some comments:

Lines 32-36: add reference

Line 38: please edit the sentence as genes do not catalyse biochemical changes! Enzymes catalyse biochemical reactions!!

Line 117: ‘a ripe kiwifruit remain green externally’. This is incorrect. You need to rephrase as ‘a ripe ‘Hayward’ kiwifruit remains green internally’ as it is the flesh that stays green and it is a characteristic of the ‘Hayward’ cultivar, a mutant that does not degreen. Not all of the Actinidia chinensis var. deliciosa genotypes have a bright emerald green as ‘Hayward’, some tend to be more of a yellow-green colour. Add reference

Lines 125-126: add citation

Lines 161-167: mention the role of transcription factors for anthocyanin synthesis: MYB and MBW complex

Lines 190-191: unclear why you have this sentence about tomato extracts and effect of pectins on viscosity

Line 233: edit to have ‘organic acids’

Line 235: is (49) the right citation here?

Line 238: (50) is the wrong citation

Line 250: is (53) right citation here?

Line 251: (45) wrong citation

Lines 431-432: needs editing as it is confusing. Not sure where the papaya gene comes from and how it fits in the context. none of the references cited are about papaya. Please check

Line 436: could you please avoid ‘cross-talk’? many high impact journals are discouraging the use of this terminology, as pathways do not really talk.

 Line 484: typo in furthermore

Lines 660-663: the reference is not accurate here, please cite the paper where miR858 and miR828 are described first.

Author Response

Dear Reviewers,

Thanks for your diligent work.

Here is a summary of our actions in response to your message, and all the revisions have been labeled in Yellow in the manuscript:

Reviewer 1

The manuscript reviews the latest research on fruit ripening in fleshy fruit. I found it comprehensive and informative, although information dense at times. A number of aspects of the physiology of fruit ripening have been considered, from metabolic changes associated with fruit composition to transcriptional regulation. The title could be more informative, e.g. Molecular and hormonal mechanisms controlling ripening in fleshy fruit.

The authors should revise section 2.3, focusing on the sugar and starch metabolism. References are not accurate, and more details should be added.

Response: The section on sugars and acids has been revised; information on starch metabolism has been added, please check section 2.3, line 270-313.

Some comments:

Lines 32-36: add reference

Response: we have restructured this sentence (line 37-38) and cited later sections of the review where further information on changes in the plastid, vacuole and nucleus are discussed.

Line 38: please edit the sentence as genes do not catalyse biochemical changes! Enzymes catalyse biochemical reactions!!

Response: We corrected the sentence, please check line 40.

Line 117: ‘a ripe kiwifruit remain green externally’. This is incorrect. You need to rephrase as ‘a ripe ‘Hayward’ kiwifruit remains green internally’ as it is the flesh that stays green and it is a characteristic of the ‘Hayward’ cultivar, a mutant that does not degreen. Not all of the Actinidia chinensis var. deliciosa genotypes have a bright emerald green as ‘Hayward’, some tend to be more of a yellow-green colour. Add reference

Response: Thanks for the comment. The original comment was intended to refer to external color change. We have altered the sentence to clarify this, added a comment about the external green/brown appearance, and commented on differences in kiwifruit internal color in two of the most common kiwifruit types, please see lines 195-202.

Lines 125-126: add citation

Response: the citation was added, please check line 210.

Lines 161-167: mention the role of transcription factors for anthocyanin synthesis: MYB and MBW complex

Response: We have added this information, please see lines 189-194.

Lines 190-191: unclear why you have this sentence about tomato extracts and effect of pectins on viscosity

Response: The viscosity result shows PG-antisense has an effect on properties of the cell wall. We have added two additional references that show small effects on softening have also been detected. Please see lines 224-226.

Line 233: edit to have ‘organic acids’

Response: We agreed and added ‘organic’ to the sub-title, please check line 268.

Line 235: is (49) the right citation here?

Response: The original citation (49) was removed, the remaining citation (previous 48 now 69) is enough to support the point here, please check line 271.

Line 238: (50) is the wrong citation

Response: The section on sugars and acids has been revised, information on starch metabolism has been added, please check section 2.3, line 268-313.

Line 250: is (53) right citation here?

Response: The section on sugars and acids has been rephrased; information on starch metabolism has been added, please check section 2.3, line 268-313.

Line 251: (45) wrong citation

Response: The section on sugars and acids has been rephrased; information on starch metabolism has been added, please check section 2.3, line 268-313.

Lines 431-432: needs editing as it is confusing. Not sure where the papaya gene comes from and how it fits in the context. none of the references cited are about papaya. Please check

Response: We removed the comment on the papaya gene here, which does not affect the point we want to express in this sentence, please check line 497.

Line 436: could you please avoid ‘cross-talk’? many high impact journals are discouraging the use of this terminology, as pathways do not really talk.

Response: We changed ‘cross talk’ to ‘interactions’ to makes it more clear, please check line 502.

 Line 484: typo in furthermore

Response: We revised the typo, thanks for pointing it out, please check line 550.

Lines 660-663: the reference is not accurate here, please cite the paper where miR858 and miR828 are described first.

Response: We have restructured this sentence and cited properly, please check line 722-725.

Reviewer 2 Report

In the present review, the authors gathered a considerable amount of information about virtually all mechanisms regulating fruit ripening. The organization of the review is quite didactic, starting from the main physiological and biochemical changes during ripening (fruit softening, carotenoid and flavonoid accumulation, synthesis of sugars, acids and volatiles that contribute to taste and aroma) and the genes and mechanisms that control them. The authors then revised the main hormones and transcriptional factor involved in fruit ripening as well as their interaction with environmental factors such as light and temperature. The epigenetic control of fruit ripening through DNA methylation, histone modification and non-coding RNAs was also included. The review is comprehensive and  goes far beyond simply compiling the existing literature. On the contrary, the authors organized the existing literature, to which they made relevant contributions, and brought novel ideas and hypotheses from it. Thus, this review represents a good contribution for the research area and it will be useful to a broad community of researchers. Below are some specific points that may contribute to the improvement of the text.

Line 15 – “Light, detected by different photoreceptors and operating through HY5”. The full name of the HY5 gene (ELONGATED HYPOCOTYL 5) should be entered the first time it is mentioned.

Line 47 – “…non-climacteric fruits (i.e. citrus fruits such as grapefruit, orange and lemon, raspberry, strawberry, etc.). This sentence should be reformulated, as it gives the impression that raspberries and strawberries are also citrus fruits.

Line 56 – “Hormone signalling operates primarily by modulating expression or action of TFs plus genes involved in hormone biosynthesis, perception and signalling. Expression of other ripening TFs is regulated developmentally or by environmental stimuli.” It would be interesting to report that hormones may also be downstream from TF. For example, the KNOX genes regulate the biosynthesis of GA and cytokinin.

Line 59 – “The tomato (Solanum lycopersicum) was selected early on as a model for fruit ripening and was the first fruit-bearing plant to have the complete genome sequence determined”. Actually, the first fruit-bearing plant to have the complete genome sequenced was Arabidopsis. The tomato is also not the first flesh fruit-bearing plant to have its genome sequenced (the grape came first), although it was the first complete genome.

Line 99 – “Visual clues are extremely important for signalling to frugivores the availability of ripe fruit and a wide range of animals, including seed-dispersers such as bats, rodents, primates,..” It would be interesting to rephrase this sentence taking into account that bats and rodents are actually color-blind, although they probably distinguish ripe fruits from unripe ones.  

Lines 106 to 110. It would be interesting to mention that, unlike carotenoids, flavonoids, including anthocyanins, normally accumulate in external tissues of the pericarp. An interesting example is the skin of tomatoes, whose accumulation of the flavonoid rutin is controlled by the MYB12 transcription factor.

Line 116 – “Color change mechanisms are not universal, however, and ripe kiwifruit remain green externally”. Actually, kiwifruit fruits are brown externally and green in their inner pericarp.

Figure 1. The information in the legend of this figure regarding different species of fruit  that accumulate specific carotenoids is very interesting and can be better explored in a table referring to the figure. A table would allow the insertion of bibliography and Latin names of exotic fruits such as gac. In addition to gac, other exotic fruits that accumulate lycopene are guava (Psidium guajava) and pitanga (Eugenia uniflora).

Figure 1. Why include only ABA, since there are other important apocarotenoid hormones in this pathway, such as strigolactones and beta-cyclocital?

Figure 2. What is the relevance of indicating tomato anthocyanin mutants in this figure? These mutants were isolated based on the accumulation of anthocyanin in the hypocotyl and not in the fruit. Although tomatoes accumulate flavonoids in the fruit skin, it is not considered a fruit that accumulates anthocyanin. Perhaps it is more informative to indicate different types of anthocyanins and examples of fruits that accumulate them, as in Fig. 1 for carotenoids.

Line 409 – “Auxin is synthesised from chorismite in the chloroplasts by metabolism to tryptophan and then via either indole-3-acetamide or indole-3-pyruvic acid, to generate IAA(118)”.  There are more recent auxin biosynthesis reviews, especially those taking into account that the main pathway for most plants is composed of only two steps based on TAA and YUCCA genes.  

Line 550 – “In 1991, differential DNA methylation was first reported in two tomato species by Messeguer et al.(166)”. There is only one species of tomato (Solanum lycopersicum). In the referred work by Messeguer et al. the other species used is not the tomato, but a wild relative named S. peruvianum (formerly Lycopersicon peruvianum as mentioned in the article).

Line 660 – “Other miRNAs also regulate fruit coloration and tomato miR828 is a negative regulator of anthocyanin accumulation in kiwifruit..”. Please clarify that kiwifruit has a miR828 homologue that is a negative regulator of anthocyanin accumulation in this species.

Line 674 to 690 should be summarized for just the introduction of LncRNAs, since most of the information placed in this paragraph is outside the scope of the review.

Line 832 – “6.3. The nor (NAC-NOR) mutation”. It could be interesting to discuss the weak NAC mutant alcobaça, which is allelic to nor.

Figure 7. Although fruit development is not the subject of this review, the proposed model gives the idea that this process is controlled only by auxin and ARFs. It would be important to include other players, such as gibberellin and cell cycle genes.

Author Response

Dear Reviewers,

Thanks for your diligent work.

Here is a summary of our actions in response to your message, and all the revisions have been labeled in Yellow in the manuscript:

Line 15 – “Light, detected by different photoreceptors and operating through HY5”. The full name of the HY5 gene (ELONGATED HYPOCOTYL 5) should be entered the first time it is mentioned.

Response: The full name of HY5 was added at first mention (the abstract), please check line 17.

Line 47 – “…non-climacteric fruits (i.e. citrus fruits such as grapefruit, orange and lemon, raspberry, strawberry, etc.). This sentence should be reformulated, as it gives the impression that raspberries and strawberries are also citrus fruits.

Response: The confusing description was removed and replaced with ‘… non-climacteric fruits(i.e. citrus, grape, orange, lemon, raspberry, strawberry, etc)’, please check it in line 49.

Line 56 – “Hormone signalling operates primarily by modulating expression or action of TFs plus genes involved in hormone biosynthesis, perception and signalling. Expression of other ripening TFs is regulated developmentally or by environmental stimuli.” It would be interesting to report that hormones may also be downstream from TF. For example, the KNOX genes regulate the biosynthesis of GA and cytokinin.

Response: We have added two sentences about the interplay between developmental, environmental and hormonal cues, pointing out that hormones can affect TFs and TFs can also affect hormones. Please check line 59-63.

Line 59 – “The tomato (Solanum lycopersicum) was selected early on as a model for fruit ripening and was the first fruit-bearing plant to have the complete genome sequence determined”. Actually, the first fruit-bearing plant to have the complete genome sequenced was Arabidopsis. The tomato is also not the first flesh fruit-bearing plant to have its genome sequenced (the grape came first), although it was the first complete genome.

Response: Thanks for the botanical point. We are referring to complete sequences of fleshy fruits; sorry for not making that clear. The sentence was revised to ‘The tomato (Solanum lycopersicum) was selected as a model for fruit ripening and the complete genome sequence was determined in 2012’, please check line 64-65.

Line 99 – “Visual clues are extremely important for signalling to frugivores the availability of ripe fruit and a wide range of animals, including seed-dispersers such as bats, rodents, primates,..” It would be interesting to rephrase this sentence taking into account that bats and rodents are actually color-blind, although they probably distinguish ripe fruits from unripe ones.  

Response: We agree that this is an interesting point, we have slightly edited this section and added 5-6 new lines. We do not wish to be too categorical about color discrimination, since old views have recently been questioned or modified. We believe that the information we have added is accurate, please check line 108-112.

Lines 106 to 110. It would be interesting to mention that, unlike carotenoids, flavonoids, including anthocyanins, normally accumulate in external tissues of the pericarp. An interesting example is the skin of tomatoes, whose accumulation of the flavonoid rutin is controlled by the MYB12 transcription factor.

Response: We have added two sentences about the description of anthocyanins and its synthesis from thephenylpropanoid pathway, please check line 168-171.

Line 116 – “Color change mechanisms are not universal, however, and ripe kiwifruit remain green externally”. Actually, kiwifruit fruits are brown externally and green in their inner pericarp.

Response: The sentence has been revised. We added comments about two of the most common kiwifruit here and described as ‘however, and ripe ‘Hayward’ and ‘Hort16A’ kiwifruit remain green and gold internally respectively’ with reference, please see lines 195-202.

Figure 1. The information in the legend of this figure regarding different species of fruit that accumulate specific carotenoids is very interesting and can be better explored in a table referring to the figure. A table would allow the insertion of bibliography and Latin names of exotic fruits such as gac. In addition to gac, other exotic fruits that accumulate lycopene are guava (Psidium guajava) and pitanga (Eugenia uniflora).

Response: All of the Latin names of each fruit species mentioned in the figure were added in the figure legend, please check line 152-159. The figure was redrawn and simplified from Lado et al. (2016) and Luan et al. (2020), a detailed table including the constituents and content of various carotenoid was given in the book chapter of Lado et al.(2016). So we decided to keep the carotenoid pathway here, which could support the ideas we discuss in this review.

Figure 1. Why include only ABA, since there are other important apocarotenoid hormones in this pathway, such as strigolactones and beta-cyclocital?

Response: We have revised the legend, with additional information. “Note that the hormone ABA, which is discussed in section 3.3, is produced from this pathway. Some of these metabolites are also utilized in the biosynthesis of hormones, such as GA, strigolactones and beta-cyclocital but these pathways are omitted from this Figure because they are not discussed in this review”, please check line 164-167.

Figure 2. What is the relevance of indicating tomato anthocyanin mutants in this figure? These mutants were isolated based on the accumulation of anthocyanin in the hypocotyl and not in the fruit. Although tomatoes accumulate flavonoids in the fruit skin, it is not considered a fruit that accumulates anthocyanin. Perhaps it is more informative to indicate different types of anthocyanins and examples of fruits that accumulate them, as in Fig. 1 for carotenoids.

Response: Thanks for the comments, the anthocyanin mutants were removed from the figure to avoid confusion. The anthocyanins are complex and variable compounds of compared to the simpler carotenoid pathway. The sentence about various anthocyanin in different fruits was revised to give more detailed information in the relevant paragraph in the main text, not in the figure legend, please check line 171-175.

Line 409 – “Auxin is synthesised from chorismite in the chloroplasts by metabolism to tryptophan and then via either indole-3-acetamide or indole-3-pyruvic acid, to generate IAA(118)”.  There are more recent auxin biosynthesis reviews, especially those taking into account that the main pathway for most plants is composed of only two steps based on TAA and YUCCA genes.  

Response: We have added reference to the Trp-independent pathway, with references, please check line 473-477.

Line 550 – “In 1991, differential DNA methylation was first reported in two tomato species by Messeguer et al.(166)”. There is only one species of tomato (Solanum lycopersicum). In the referred work by Messeguer et al. the other species used is not the tomato, but a wild relative named S. peruvianum (formerly Lycopersicon peruvianum as mentioned in the article).

Response: The sentence was revised to ‘DNA methylation was first reported in two tomato genera by Messeguer et al.’, Please check line 620.

Line 660 – “Other miRNAs also regulate fruit coloration and tomato miR828 is a negative regulator of anthocyanin accumulation in kiwifruit.”. Please clarify that kiwifruit has a miR828 homologue that is a negative regulator of anthocyanin accumulation in this species.

Response: We have restructured this sentence and cited properly, please check line 722-725.

Line 674 to 690 should be summarized for just the introduction of LncRNAs, since most of the information placed in this paragraph is outside the scope of the review.

Response: This paragraph has been shortened, focusing more on lncRNA and functions in fruits, please check line 734-753.

Line 832 – “6.3. The nor (NAC-NOR) mutation”. It could be interesting to discuss the weak NAC mutant alcobaça, which is allelic to nor.

Response: Although the gene mapping and physiology is clear, we feel there is insufficient molecular information about this to warrant inclusion.

Figure 7. Although fruit development is not the subject of this review, the proposed model gives the idea that this process is controlled only by auxin and ARFs. It would be important to include other players, such as gibberellin and cell cycle genes.

Response: We decline to make any change: We feel that GA and development is beyond the scope of this review. Also, in tomato, which is an important focus, cell division ceases after approximately 2 weeks of development, several works before the initiation of ripening.

Reviewer 3 Report

The manuscript entitled "Fruit Ripening Mechanisms" by Li et al. was reviewed for publication in Cells (manuscript cells-1189703). The manuscript, coauthored (corresponding author) by one of the most well-known experts in the field, thoroughly reviews current knowledge about fleshy fruit ripening at the molecular level, synthesizes knowledge acquired from the main model (tomato) for fleshy fruit research, and in addition discusses progress with other non-model crop species. Furthermore, it addresses and discusses recent research that has forced a re-evaluation of the “master” transcription factor roles in the control of fleshy fruit ripening. The manuscript is very well written, easy to read and full of up to date information about this very important and well researched topic of wide interest.

Comments and Corrections

Carotenoids Section, I think it would be important to mention in this section that fleshy fruits are good sources of provitamin A carotenoids, hence the importance of this pathway for human health.

I found several phrases that were difficult to understand, so I indicate them here and directly on the pdf I uploaded to facilitate the revisions:

Lines 121-123, “and some thylakoids” is unclear. The retain some thylakoids. Please clarify.

Lines 470-472, “can kiwifruit ripening” does not make sense.

Line 582, what is meant by “newer”? Please clarify.

Line 596, should be carotenoid metabolism?

Line 695, “attribute by…”, please clarify.

I made minor correction suggestions directly on the manuscript pdf file.

Author Response

Dear Reviewers,

Thanks for your diligent work.

Here is a summary of our actions in response to your message, and all the revisions have been labeled in Yellow in the manuscript:

Comments and Corrections

Carotenoids Section, I think it would be important to mention in this section that fleshy fruits are good sources of provitamin A carotenoids, hence the importance of this pathway for human health.

Response: we agree and a sentence including citation has been added about this (line 130).

I found several phrases that were difficult to understand, so I indicate them here and directly on the pdf I uploaded to facilitate the revisions:

Lines 121-123, “and some thylakoids” is unclear. The retain some thylakoids. Please clarify.

Response: The sentence was revised and now reads: Tomatoes normally lose chlorophyll and accumulate carotenoids, but “greenflesh” mutant fruit retain some thylakoids and chlorophylls and at the same time they accumulate carotenoids and consequently the fruit appear a dirty-brown color’, please check line 205-209.

Lines 470-472, “can kiwifruit ripening” does not make sense.

Response: The sentence was revised to ‘Wu et al. showed that external ethylene and MeJA increased ethylene production that was correlated with higher transcripts of ACS genes AdACS1 and AdACS2 and ACS enzyme activity in kiwifruit’, please check from line 536-538.

Line 582, what is meant by “newer”? Please clarify.

Response: We removed ‘newer’, to avoid confusion in this sentence, please check line 650.

Line 596, should be carotenoid metabolism?

Response: The sentence was revised with the ‘metabolism’ added, ‘Tomato proteins HP1 and HP2 are involved in fruit carotenoids metabolism’, please check line 664.

Line 695, “attribute by…”, please clarify.

Response: This paragraph has been shortened and this sentence has been removed, please check line 734-753.

I made minor correction suggestions directly on the manuscript pdf file.

Response: Thanks for the corrections, all the minor revisions labeled are accepted.
